# CONVERGENT ADAPTIVE GRADIENT METHODS IN DECENTRALIZED OPTIMIZATION

## ABSTRACT

Adaptive gradient methods including Adam, AdaGrad, and their variants have been very successful for training deep learning models, such as neural networks, in the past few years. Meanwhile, given the need for distributed training procedures, distributed optimization algorithms are at the center of attention. With the growth of computing power and the need for using machine learning models on mobile devices, the communication cost of distributed training algorithms needs careful consideration. In that regard, more and more attention is shifted from the traditional parameter server training paradigm to the decentralized one, which usually requires lower communication costs. In this paper, we rigorously incorporate adaptive gradient methods into decentralized training procedures and introduce novel convergent decentralized adaptive gradient methods. Specifically, we propose a general algorithmic framework that can convert existing adaptive gradient methods to their decentralized counterparts. In addition, we thoroughly analyze the convergence behavior of the proposed algorithmic framework and show that if a given adaptive gradient method converges, under some specific conditions, then its decentralized counterpart is also convergent.

## 1 INTRODUCTION

Distributed training of machine learning models is drawing growing attention in the past few years due to its practical benefits and necessities. Given the evolution of computing capabilities of CPUs and GPUs, computation time in distributed settings is gradually dominated by the communication time in many circumstances (Chilimbi et al., 2014; McMahan et al., 2017). As a result, a large amount of recent works has been focussing on reducing communication cost for distributed learning (Alistarh et al., 2017; Lin et al., 2018; Wangni et al., 2018; Stich et al., 2018; Wang et al., 2018; Tang et al., 2019). In the traditional parameter (central) server setting, where a parameter server is employed to manage communication in the whole network, many effective communication reductions have been proposed based on gradient compression (Aji & Heafield, 2017) and quantization (Chen et al., 2010; Ge et al., 2013; Jegou et al., 2010) techniques. Despite these communication reduction techniques, its cost still, usually, scales linearly with the number of workers. Due to this limitation and with the sheer size of decentralized devices, the *decentralized training paradigm* (Duchi et al., 2011b), where the parameter server is removed and each node only communicates with its neighbors, is drawing attention. It has been shown in Lian et al. (2017) that decentralized training algorithms can outperform parameter server-based algorithms when the training bottleneck is the communication cost. The decentralized paradigm is also preferred when a central parameter server is not available.

In light of recent advances in nonconvex optimization, an effective way to accelerate training is by using adaptive gradient methods like AdaGrad (Duchi et al., 2011a), Adam (Kingma & Ba, 2015) or AMSGrad (Reddi et al., 2018). Their popularity are due to their practical benefits in training neural networks, featured by faster convergence and ease of parameter tuning compared with Stochastic Gradient Descent (SGD) (Robbins & Monro, 1951). Despite a large amount of studies within the distributed optimization literature, few works have considered bringing adaptive gradient methods into distributed training, largely due to the lack of understanding of their convergence behaviors. Notably, Reddi et al. (2020) develop the first decentralized ADAM method for distributed optimization problems with a direct application to federated learning. An inner loop is employed to compute mini-batch gradients on each node and a global adaptive step is applied to update the global parameter at each outer iteration. Yet, in the settings of our paper, nodes can only communicate *to their neighbors* on a fixed communication graph while a server/worker communication is required in Reddi et al. (2020). Designing adaptive methods in such settings is highly non-trivial due to the already complex update rules and to the interaction between the effect of using adaptive learning

rates and the decentralized communication protocols. This paper is an attempt at bridging the gap between both realms in nonconvex optimization. Our contributions are summarized as follows:

- In this paper, we investigate the possibility of using adaptive gradient methods in the decentralized training paradigm, where nodes have only a local view of the whole communication graph. We develop a general technique that converts an adaptive gradient method from a centralized method to its decentralized variant.

- By using our proposed technique, we present a new decentralized optimization algorithm, called decentralized AMSGrad, as the decentralized counterpart of AMSGrad.

- We provide a theoretical verification interface, in Theroem 2, for analyzing the behavior of decentralized adaptive gradient methods obtained as a result of our technique. Thus, we characterize the convergence rate of decentralized AMSGrad, which is the first convergent decentralized adaptive gradient method, to the best of our knowledge.

A *novel technique* in our framework is a mechanism to enforce a *consensus on adaptive learning rates* at different nodes. We show the importance of consensus on adaptive learning rates by proving a divergent problem instance for a recently proposed decentralized adaptive gradient method, namely DADAM (Nazari et al., 2019), a decentralized version of AMSGrad. Though consensus is performed on the model parameter, DADAM lacks consensus principles on adaptive learning rates.

After having presented existing related work and important concepts of decentralized adaptive methods in Section 2, we develop our general framework for converting any adaptive gradient algorithm in its decentralized counterpart along with their rigorous finite-time convergence analysis in Section 3 concluded by some illustrative examples of our framework's behavior in practice.

**Notations**: $x_{t,i}$ denotes variable $x$ at node $i$ and iteration $t$. $\|\cdot\|_{abs}$ denotes the entry-wise $L_1$ norm of a matrix, i.e. $\|A\|_{abs} = \sum_{i,j} |A_{i,j}|$. We introduce important notations used throughout the paper: for any $t > 0$, $G_t := [g_{t,N}]$ where $[g_{t,N}]$ denotes the matrix $[g_{t,1}, g_{t,2}, \cdots, g_{t,N}]$ (where $g_{t,i}$ is a column vector), $M_t := [m_{t,N}]$, $X_t := [x_{t,N}]$, $\overline{\nabla f}(X_t) := \frac{1}{N} \sum_{i=1}^{N} \nabla f_i(x_{t,i})$, $U_t := [u_{t,N}]$, $\tilde{U}_t := [\tilde{u}_{t,N}]$, $V_t := [v_{t,N}]$, $\hat{V}_t := [\hat{v}_{t,N}]$, $\overline{X}_t := \frac{1}{N} \sum_{i=1}^{N} x_{t,i}$, $\overline{U}_t := \frac{1}{N} \sum_{i=1}^{N} u_{t,i}$ and $\overline{\tilde{U}}_t := \frac{1}{N} \sum_{i=1}^{N} \tilde{u}_{t,i}$.

## 2 DECENTRALIZED ADAPTIVE TRAINING AND DIVERGENCE OF DADAM

### 2.1 RELATED WORK

**Decentralized optimization:** Traditional decentralized optimization methods include well-know algorithms such as ADMM (Boyd et al., 2011), Dual Averaging (Duchi et al., 2011b), Distributed Subgradient Descent (Nedic & Ozdaglar, 2009). More recent algorithms include Extra (Shi et al., 2015), Next (Di Lorenzo & Scutari, 2016), Prox-PDA (Hong et al., 2017), GNSD (Lu et al., 2019), and Choco-SGD (Koloskova et al., 2019). While these algorithms are commonly used in applications other than deep learning, recent algorithmic advances in the machine learning community have shown that decentralized optimization can also be useful for training deep models such as neural networks. Lian et al. (2017) demonstrate that a stochastic version of Decentralized Subgradient Descent can outperform parameter server-based algorithms when the communication cost is high. Tang et al. (2018) propose the D$^2$ algorithm improving the convergence rate over Stochastic Subgradient Descent. Assran et al. (2019) propose the Stochastic Gradient Push that is more robust to network failures for training neural networks. The study of decentralized training algorithms in the machine learning community is only at its initial stage. No existing work, to our knowledge, has seriously considered integrating *adaptive gradient methods* in the setting of decentralized learning. One noteworthy work (Nazari et al., 2019) propose a decentralized version of AMSGrad (Reddi et al., 2018) and it is proven to satisfy some non-standard regret.

**Adaptive gradient methods:** Adaptive gradient methods have been popular in recent years due to their superior performance in training neural networks. Most commonly used adaptive methods include AdaGrad (Duchi et al., 2011a) or Adam (Kingma & Ba, 2015) and their variants. Key features of such methods lie in the use of momentum and adaptive learning rates (which means that the learning rate is changing during the optimization and is anisotropic, i.e. depends on the dimension). The method of reference, called Adam, has been analyzed in Reddi et al. (2018) where the authors point out an error in previous convergence analyses. Since then, a variety of papers have

been focussing on analyzing the convergence behavior of the numerous existing adaptive gradient methods. Ward et al. (2019), Li & Orabona (2019) derive convergence guarantees for a variant of AdaGrad without coordinate-wise learning rates. Chen et al. (2019) analyze the convergence behavior of a broad class of algorithms including AMSGrad and AdaGrad. Zou & Shen (2018) provide a unified convergence analysis for AdaGrad with momentum. Noticeable recent works on adaptive gradient methods can be found in Agarwal et al. (2019); Luo et al. (2019); Zaheer et al. (2018).

## 2.2 DECENTRALIZED OPTIMIZATION

In distributed optimization (with $N$ nodes), we aim at solving the following problem

$$\min_{x \in \mathbb{R}^d} \frac{1}{N} \sum_{i=1}^{N} f_i(x), \tag{1}$$

where $x$ is the vector of parameters and $f_i$ is only accessible by the $i$th node. Through the prism of empirical risk minimization procedures, $f_i$ can be viewed as the average loss of the data samples located at node $i$, for all $i \in [N]$. Throughout the paper, we make the following mild assumptions required for analyzing the convergence behavior of the different decentralized optimization algorithms:

**A1.** *For all $i \in [N]$, $f_i$ is differentiable and the gradients is $L$-Lipschitz, i.e., for all $(x, y) \in \mathbb{R}^d$, $\|\nabla f_i(x) - \nabla f_i(y)\| \leq L\|x - y\|$.*

**A2.** *We assume that, at iteration $t$, node $i$ accesses a stochastic gradient $g_{t,i}$. The stochastic gradients and the gradients of $f_i$ have bounded $L_\infty$ norms, i.e. $\|g_{t,i}\| \leq G_\infty$, $\|\nabla f_i(x)\|_\infty \leq G_\infty$.*

**A3.** *The gradient estimators are unbiased and each coordinate have bounded variance, i.e. $\mathbb{E}[g_{t,i}] = \nabla f_i(x_{t,i})$ and $\mathbb{E}[([g_{t,i} - f_i(x_{t,i})]_j)^2] \leq \sigma^2, \forall t, i, j$.*

Assumptions A1 and A3 are standard in distributed optimization literature. A2 is slightly stronger than the traditional assumption that the estimator has bounded variance, but is commonly used for the analysis of adaptive gradient methods (Chen et al., 2019; Ward et al., 2019). Note that the bounded gradient estimator assumption in A2 implies the bounded variance assumption in A3. In decentralized optimization, the nodes are connected as a graph and each node only communicates to its neighbors. In such case, one usually constructs a $N \times N$ matrix $W$ for information sharing when designing new algorithms. We denote $\lambda_i$ to be its $i$th largest eigenvalue and define $\lambda \triangleq \max(|\lambda_2|, |\lambda_N|)$. The matrix $W$ cannot be arbitrary, its required key properties are listed in the following assumption:

**A4.** *The matrix $W$ satisfies:* (I) $\sum_{j=1}^{N} W_{i,j} = 1$, $\sum_{i=1}^{N} W_{i,j} = 1$, $W_{i,j} \geq 0$, (II) $\lambda_1 = 1$, $|\lambda_2| < 1$, $|\lambda_N| < 1$ *and* (III) $W_{i,j} = 0$ *if node $i$ and node $j$ are not neighbors.*

We now present the failure to converge of current decentralized adaptive method before introducing our proposed framework for general decentralized adaptive gradient methods.

## 2.3 DIVERGENCE OF DADAM

Recently, Nazari et al. (2019) initiated an attempt to bring adaptive gradient methods into decentralized optimization with Decentralized ADAM (DADAM), shown in Algorithm 1. DADAM is essentially a decentralized version of ADAM and the key modification is the use of a consensus step on the optimization variable $x$ to transmit information across the network, encouraging its convergence. The matrix $W$ is a doubly stochastic matrix (which satisfies A4) for achieving average consensus of $x$. Introducing such mixing matrix is standard for decentralizing an algorithm, such as distributed gradient descent (Nedic & Ozdaglar, 2009; Yuan et al.,

---

**Algorithm 1** DADAM (with N nodes)

1: **Input:** $\alpha$, current point $X_t$, $u_{\frac{1}{2},i} = \hat{v}_{0,i} = \epsilon \mathbf{1}$, $m_0 = 0$ and mixing matrix $W$
2: **for** $t = 1, 2, \cdots, T$ **do**
3:     **for all** $i \in [N]$ **do in parallel**
4:         $g_{t,i} \leftarrow \nabla f_i(x_{t,i}) + \xi_{t,i}$
5:         $m_{t,i} = \beta_1 m_{t-1,i} + (1 - \beta_1)g_{t,i}$
6:         $v_{t,i} = \beta_2 v_{t-1,i} + (1 - \beta_2)g_{t,i}^2$
7:         $\hat{v}_{t,i} = \beta_3 \hat{v}_{t,i} + (1 - \beta_3)\max(\hat{v}_{t-1,i}, v_{t,i})$
8:         $x_{t+\frac{1}{2},i} = \sum_{j=1}^{N} W_{ij} x_{t,j}$
9:         $x_{t+1,i} = x_{t+\frac{1}{2},i} - \alpha \frac{m_{t,i}}{\sqrt{\hat{v}_{t,i}}}$
10: **end for**

---

2016). It is proven in Nazari et al. (2019) that DADAM admits a non-standard regret bound in the online setting. Nevertheless, whether the algorithm can converge to stationary points in standard offline settings such training neural networks is still unknown. The next theorem shows that DADAM may fail to converge in the offline settings.

**Theorem 1.** *There exists a problem satisfying A1-A4 where DADAM fails to converge to a stationary points with $\nabla f(\bar{X}_t) = 0$.*

*Proof.* Consider a two-node setting with objective function $f(x) = 1/2 \sum_{i=1}^{2} f_i(x)$ and $f_1(x) = \mathbb{1}[|x| \le 1]2x^2 + \mathbb{1}[|x| > 1](4|x| - 2)$, $f_2(x) = \mathbb{1}[|x-1| \le 1](x-1)^2 + \mathbb{1}[|x-1| > 1](2|x-1| - 1)$. We set the mixing matrix $W = [0.5, 0.5; 0.5, 0.5]$. The optimal solution is $x^* = 1/3$. Both $f_1$ and $f_2$ are smooth and convex with bounded gradient norm 4 and 2, respectively. We also have $L = 4$ (defined in A1). If we initialize with $x_{1,1} = x_{1,2} = -1$ and run DADAM with $\beta_1 = \beta_2 = \beta_3 = 0$ and $\epsilon \le 1$, we will get $\hat{v}_{1,1} = 16$ and $\hat{v}_{1,2} = 4$. Since $|g_{t,1}| \le 4, |g_{t,2}| \le 2$ due to bounded gradient, and $(\hat{v}_{t,1}, \hat{v}_{t,2})$ are non-decreasing, we have $\hat{v}_{t,1} = 16, \hat{v}_{t,2} = 4, \forall t \ge 1$. Thus, after $t = 1$, DADAM is equivalent to running decentralized gradient descent (DGD) (Yuan et al., 2016) with a re-scaled $f_1$ and $f_2$, *i.e.* running DGD on $f'(x) = \sum_{i=1}^{2} f_i'(x)$ with $f_1'(x) = 0.25f_1(x)$ and $f_2'(x) = 0.5f_2(x)$, which unique optimal $x' = 0.5$. Define $\bar{x}_t = (x_{t,1} + x_{t,2})/2$, then by Th. 2 in Yuan et al. (2016), we have when $\alpha < 1/4$, $f'(\bar{x}_t) - f(x') = O(1/(\alpha t))$. Since $f'$ has a unique optima $x'$, the above bound implies $\bar{x}_t$ is converging to $x' = 0.5$ which has non-zero gradient on function $\nabla f(0.5) = 0.5$. $\square$

Theorem 1 shows that, even though DADAM is proven to satisfy some regret bounds (Nazari et al., 2019), it can fail to converge to stationary points in the nonconvex offline setting (common for training neural networks). We conjecture that this inconsistency in the convergence behavior of DADAM is due to the definition of the regret in Nazari et al. (2019). The next section presents decentralized adaptive gradient methods that are guaranteed to converge to stationary points under assumptions and provide a characterization of that convergence in finite-time and independently of the initialization.

## 3 CONVERGENCE OF DECENTRALIZED ADAPTIVE GRADIENT METHODS

In this section, we discuss the difficulties of designing adaptive gradient methods in decentralized optimization and introduce an algorithmic framework that can turn some existing convergent adaptive gradient methods to their decentralized counterparts. We also develop the first convergent decentralized adaptive gradient method, converted from AMSGrad, as an instance of this framework.

### 3.1 IMPORTANCE AND DIFFICULTIES OF CONSENSUS ON ADAPTIVE LEARNING RATES

The divergent example in the previous section implies that we should synchronize the adaptive learning rates on different nodes. This can be easily achieved in the parameter server setting where all the nodes are sending their gradients to a central server at each iteration. The parameter server can then exploit the received gradients to maintain a sequence of synchronized adaptive learning rates when updating the parameters, see Reddi et al. (2020). However, in our decentralized setting, every node can only communicate with its neighbors and such central server does not exist. Under that setting, the information for updating the adaptive learning rates can only be shared locally instead of broadcasted over the whole network. This makes it impossible to obtain, in a single iteration, a synchronized adaptive learning rate update using all the information in the network.

*Systemic Approach:* On a systemic level, one way to alleviate this bottleneck is to design communication protocols in order to give each node access to the same aggregated gradients over the whole network, at least periodically if not at every iteration. Therefore, the nodes can update their individual adaptive learning rates based on the same shared information. However, such solution may introduce an extra communication cost since it involves broadcasting the information over the whole network.

*Algorithmic Approach:* Our contributions being on an algorithmic level, another way to solve the aforementioned problem is by letting the sequences of adaptive learning rates, present on different nodes, to gradually *consent*, through the iterations. Intuitively, if the adaptive learning rates can consent fast enough, the difference among the adaptive learning rates on different nodes will not affect the convergence behavior of the algorithm. Consequently, no extra communication costs need to be introduced. We now develop this exact idea within the existing adaptive methods stressing on the need for a relatively low-cost and easy-to-implement consensus of adaptive learning rates.

### 3.2 DECENTRALIZED ADAPTIVE GRADIENT UNIFYING FRAMEWORK

We now develop a method that implements consensus of adaptive learning rates. While each node can have different $\hat{v}_{t,i}$ in DADAM (Algorithm 1), one can keep track of the min/max/average of these adaptive learning rates and use that quantity as the new adaptive learning rate.

The predefinition of some convergent lower and upper bounds may also lead to a gradual synchronization of the adaptive learning rates on different nodes as developed for AdaBound in Luo et al. (2019). In this paper, we present an algorithm framework for decentralized adaptive gradient methods as Algorithm 2, which uses average consensus of $\hat{v}_{t,i}$ (see consensus update in line 8 and 11) to help convergence. Algorithm 2 can become different adaptive gradient methods by specifying $r_t$ as different functions. E.g., when we choose $\hat{v}_{t,i} = \frac{1}{t}\sum_{k=1}^{t} g_{k,i}^2$, Algorithm 2 becomes a decentralized version of AdaGrad. When one chooses $\hat{v}_{t,i}$ to be the adaptive learning rate for AMSGrad, we get decentralized AMSGrad (Algorithm 3). The intuition of using average consensus is that for adaptive gradient methods such as AdaGrad or Adam,

---

**Algorithm 2** Decentralized Adaptive Gradient Method (with N nodes)

---

1: **Input:** $\alpha$, initial point $x_{1,i} = x_{init}, u_{\frac{1}{2},i} = \hat{v}_{0,i}, m_{0,i} = 0$, mixing matrix $W$
2: **for** $t = 1, 2, \cdots, T$ **do**
3:    **for all** $i \in [N]$ **do in parallel**
4:       $g_{t,i} \leftarrow \nabla f_i(x_{t,i}) + \xi_{t,i}$
5:       $m_{t,i} = \beta_1 m_{t-1,i} + (1 - \beta_1)g_{t,i}$
6:       $\hat{v}_{t,i} = r_t(g_{1,i}, \cdots, g_{t,i})$
7:       $x_{t+\frac{1}{2},i} = \sum_{j=1}^{N} W_{ij} x_{t,j}$
8:       $\tilde{u}_{t,i} = \sum_{j=1}^{N} W_{ij} \tilde{u}_{t-\frac{1}{2},j}$
9:       $u_{t,i} = \max(\tilde{u}_{t,i}, \epsilon)$
10:      $x_{t+1,i} = x_{t+\frac{1}{2},i} - \alpha \frac{m_{t,i}}{\sqrt{u_{t,i}}}$
11:      $\tilde{u}_{t+\frac{1}{2},i} = \tilde{u}_{t,i} - \hat{v}_{t-1,i} + \hat{v}_{t,i}$
12: **end for**

---

$\hat{v}_{t,i}$ approximates the second moment of the gradient estimator, the average of the estimations of those second moments from different nodes is an estimation of second moment on the whole network. Also, this design will not introduce any extra hyperparameters that can potentially complicate the tuning process ($\epsilon$ in line 9 is important for numerical stability as in vanilla Adam). The main convergence result for the framework Algorithm 2 reads as follows:

**Theorem 2.** *Assume A1-A4. When* $\alpha \leq \frac{\epsilon^{0.5}}{16L}$, *Algorithm 2 yields the following regret bound*

$$\frac{1}{T}\sum_{t=1}^{T}\mathbb{E}\left[\left\|\frac{\nabla f(\overline{X}_t)}{\overline{U}_t^{1/4}}\right\|^2\right] \leq C_1\left(\frac{1}{T\alpha}(\mathbb{E}[f(Z_1)] - \min_x f(x)) + \alpha\frac{d\sigma^2}{N}\right)$$

$$+ C_2\alpha^2 d + C_3\alpha^3 d + \frac{1}{T\sqrt{N}}(C_4 + C_5\alpha)\mathbb{E}\left[\sum_{t=1}^{T}\|(-\hat{V}_{t-2} + \hat{V}_{t-1})\|_{abs}\right]$$

$$\tag{2}$$

*where* $\|\cdot\|_{abs}$ *denotes the entry-wise $L_1$ norm of a matrix (i.e $\|A\|_{abs} = \sum_{i,j}|A_{ij}|$). The constants* $C_1 = \max(4, 4L/\epsilon)$, $C_2 = 6((\beta_1/(1-\beta_1))^2 + 1/(1-\lambda)^2)LG_\infty^2/\epsilon^{1.5}$, $C_3 = 16L^2(1-\lambda)G_\infty^2/\epsilon^2$, $C_4 = 2/(\epsilon^{1.5}(1-\lambda))(\lambda + \beta_1/(1-\beta_1))G_\infty^2$, $C_5 = 2/(\epsilon^2(1-\lambda))L(\lambda + \beta_1/(1-\beta_1))G_\infty^2 + 4/(\epsilon^2(1-\lambda))LG_\infty^2$ *are independent of d, T and N. In addition,* $\frac{1}{N}\sum_{i=1}^{N}\|x_{t,i} - \overline{X}_t\|^2 \leq \alpha^2\left(\frac{1}{1-\lambda}\right)^2 dG_\infty^2\frac{1}{\epsilon}$ *which quantifies the consensus error.*

Theorem 2 shows how the convergence guarantee is affected by different factors. In addition, one can specify $\alpha$ to show convergence in terms of $T$, $d$, and $N$. An immediate result is by setting $\alpha = \sqrt{N}/\sqrt{Td}$, which is shown in Corollary 2.1.

**Corollary 2.1.** *Assume A1-A4. Set* $\alpha = \sqrt{N}/\sqrt{Td}$. *When* $\alpha \leq \frac{\epsilon^{0.5}}{16L}$, *Algorithm 2 yields the following regret bound*

$$\frac{1}{T}\sum_{t=1}^{T}\mathbb{E}\left[\left\|\frac{\nabla f(\overline{X}_t)}{\overline{U}_t^{1/4}}\right\|^2\right] \leq C_1\frac{\sqrt{d}}{\sqrt{TN}}\left((\mathbb{E}[f(Z_1)] - \min_x f(x)) + \sigma^2\right) + C_2\frac{N}{T} + C_3\frac{N^{1.5}}{T^{1.5}d^{0.5}}$$

$$+ \left(C_4\frac{1}{T\sqrt{N}} + C_5\frac{1}{T^{1.5}d^{0.5}}\right)\mathbb{E}\left[\mathcal{V}_T\right]$$

$$\tag{3}$$

*where* $\mathcal{V}_T := \sum_{t=1}^{T}\|(-\hat{V}_{t-2} + \hat{V}_{t-1})\|_{abs}$ *and* $C_1, C_2, C_3, C_4, C_5$ *are defined in Theorem 2.*

Corollary 2.1 shows that if $\mathbb{E}[\mathcal{V}_T] = o(T)$ and $\overline{U}_t$ is upper bounded, then Algorithm 2 is guaranteed to converge to stationary points of the loss function. Intuitively, this means that if the adaptive learning rates on different nodes do not change too fast, the algorithm can converge. It is shown in Chen et al. (2019) that such a condition can be satisfied by AdaGrad and AMSGrad. Besides, if this condition is

violated, the algorithm may diverge (e.g., Adam). Later, we will show convergence of decentralized versions of AMSGrad and AdaGrad by bounding this term as $O(Nd)$ and $O(Nd\log(T))$, respectively. The intuition $\mathbb{E}[\mathcal{V}_T] = o(T)$ can guarantee divergence is that the correlation between $\hat{v}_{t,i}$ and $m_{t,i}$ (due to their shared dependency on historical gradients) can make update direction negatively correlated with true gradient in expectation, leading to a non-negligible bias in updates. However, the total bias across $T$ iterations introduced by such a correlation is bounded by the term $\mathbb{E}[\mathcal{V}_T]$. Thus, if $\mathbb{E}[\mathcal{V}_T]$ grows sublinearly with $T$, convergence can still be guaranteed. Furthermore, Corollary 2.1 conveys the benefits of using more nodes in the graph. When $T$ is large enough such that the term $O(\sqrt{N}/\sqrt{Td})$ dominates the RHS of (3), linear speedup can be achieved by increasing $N$.

We now present, in Algorithm 3, a notable special case of our algorithmic framework, namely Decentralized AMSGrad, which is a decentralized variant of AMSGrad. Compared with DADAM, the above algorithm exhibits a dynamic average consensus mechanism to keep track of the average of $\{\hat{v}_{t,i}\}_{i=1}^N$, stored as $\tilde{u}_{t,i}$ on $i$th node, and uses $u_{t,i} := \max(\tilde{u}_{t,i}, \epsilon)$ for updating the adaptive learning rate for $i$th node. As the number of iteration grows, even though $\hat{v}_{t,i}$ on different nodes can converge to different constants, the $u_{t,i}$ will converge to the same number $\lim_{t\to\infty} \frac{1}{N}\sum_{i=1}^N \hat{v}_{t,i}$ if the limit exists. This average consensus mechanism enables the consensus of adaptive learning rates on different nodes, which accordingly guarantees the convergence of the method to stationary points. The consensus of adaptive learning rates is the key difference between decentralized AMSGrad and DADAM and is the reason why decentralized AMSGrad is convergent while DADAM is not.

One may notice that decentralized AMSGrad does not reduce to AMSGrad for $N = 1$ since the quantity $u_{t,i}$ in line 10 is calculated based on $v_{t-1,i}$ instead of $v_{t,i}$. This design encourages the execution of gradient computation and communication in a parallel manner. Specifically, line 4-7 (line 4-6) in Algorithm 3 (Algorithm 2) can be executed in parallel with line 8-9 (line 7-8) to overlap communication and computation time. If $u_{t,i}$ depends on $v_{t,i}$ which in turn depends on $g_{t,i}$, the gradient computation must finish before the consensus step of the adaptive learning rate in line 9. This can slow down the running time per-iteration of the algorithm. To avoid such delayed adaptive learning, adding $\tilde{u}_{t-\frac{1}{2},i} = \tilde{u}_{t,i} - \hat{v}_{t-1,i} + \hat{v}_{t,i}$ before line 9 and get rid of line 12 in Algorithm 2 is an option. Similar convergence guarantees will hold since one can easily modify our proof of Theorem 2 for such update rule. As stated above, Algorithm 3 converges, with the following rate:

---

**Algorithm 3** Decentralized AMSGrad (with N nodes)

1: **Input:** learning rate $\alpha$, initial point $x_{1,i} = x_{init}, u_{\frac{1}{2},i} = \hat{v}_{0,i} = \epsilon\mathbf{1}$ (with $\epsilon \geq 0$), $m_{0,i} = 0$, mixing matrix $W$
2: **for** $t = 1, 2, \cdots, T$ **do**
3:     **for all** $i \in [N]$ **do in parallel**
4:       $g_{t,i} \leftarrow \nabla f_i(x_{t,i}) + \xi_{t,i}$
5:       $m_{t,i} = \beta_1 m_{t-1,i} + (1-\beta_1)g_{t,i}$
6:       $v_{t,i} = \beta_2 v_{t-1,i} + (1-\beta_2)g_{t,i}^2$
7:       $\hat{v}_{t,i} = \max(\hat{v}_{t-1,i}, v_{t,i})$
8:       $x_{t+\frac{1}{2},i} = \sum_{j=1}^N W_{ij} x_{t,j}$
9:       $\tilde{u}_{t,i} = \sum_{j=1}^N W_{ij} \tilde{u}_{t-\frac{1}{2},j}$
10:      $u_{t,i} = \max(\tilde{u}_{t,i}, \epsilon)$
11:      $x_{t+1,i} = x_{t+\frac{1}{2},i} - \alpha\frac{m_{t,i}}{\sqrt{u_{t,i}}}$
12:      $\tilde{u}_{t+\frac{1}{2},i} = \tilde{u}_{t,i} - \hat{v}_{t-1,i} + \hat{v}_{t,i}$
13: **end for**

---

**Theorem 3.** *Assume A1-A4. Set $\alpha = 1/\sqrt{Td}$. When $\alpha \leq \frac{\epsilon^{0.5}}{16L}$, Algorithm 3 yields the following regret bound*

$$\frac{1}{T}\sum_{t=1}^T \mathbb{E}\left[\left\|\frac{\nabla f(\overline{X}_t)}{\overline{U}_t^{1/4}}\right\|^2\right] \leq C_1'\frac{\sqrt{d}}{\sqrt{TN}}(D_f + \sigma^2) + C_2'\frac{N}{T} + C_3'\frac{N^{1.5}}{T^{1.5}d^{0.5}} + C_4'\frac{\sqrt{N}d}{T} + C_5'\frac{Nd^{0.5}}{T^{1.5}}$$

*where $D_f := \mathbb{E}[f(Z_1)] - \min_x f(x)$, $C_1' = C_1$, $C_2' = C_2$, $C_3' = C_3$, $C_4' = C_4 G_\infty^2$ and $C_5' = C_5 G_\infty^2$. $C_1, C_2, C_3, C_4, C_5$ are constants independent of $d$, $T$ and $N$ defined in Theorem 2. In addition, the consensus of variables at different nodes is given by $\frac{1}{N}\sum_{i=1}^N \left\|x_{t,i} - \overline{X}_t\right\|^2 \leq \frac{N}{T}\left(\frac{1}{1-\lambda}\right)^2 G_\infty^2 \frac{1}{\epsilon}$.*

Theorem 3 shows that Algorithm 3 converges with a rate of $\mathcal{O}(\sqrt{d}/\sqrt{T})$ when $T$ is large, which is the best known convergence rate under the given assumptions. Note that in some related works, SGD admits a convergence rate of $\mathcal{O}(1/\sqrt{T})$ without any dependence on the dimension of the problem. Such improved convergence rate is derived under the assumption that the gradient estimator have a bounded $L_2$ norm, which can thus hide a dependency of $\sqrt{d}$ in the final convergence rate. Another

remark is the convergence measure can be converted to $\frac{1}{T}\sum_{t=1}^{T}\mathbb{E}\left[\left\|\nabla f(\overline{X}_t)\right\|^2\right]$ using the fact that $\|\overline{U}_t\|_\infty \leq G_\infty^2$ (by update rule of Algorithm 3), for the ease of comparison with existing literature.

### 3.3 CONVERGENCE ANALYSIS

The detailed proofs of this section are reported in the supplementary material.

**Proof of Theorem 2:** We now present a proof sketch for out main convergence result of Algorithm 2. *Step 1: Reparameterization.* Similarly to Yan et al. (2018); Chen et al. (2019) with SGD (with momentum) and centralized adaptive gradient methods, define the following auxiliary sequence:

$$Z_t = \overline{X}_t + \frac{\beta_1}{1-\beta_1}(\overline{X}_t - \overline{X}_{t-1}), \tag{4}$$

with $\overline{X}_0 \triangleq \overline{X}_1$. Such an auxiliary sequence can help us deal with the bias brought by the momentum and simplifies the convergence analysis. An intermediary result needed to conduct our proof reads:

**Lemma 1.** *For the sequence defined in* (4)*, we have*

$$Z_{t+1} - Z_t = \alpha \frac{\beta_1}{1-\beta_1} \frac{1}{N} \sum_{i=1}^{N} m_{t-1,i} \odot \left(\frac{1}{\sqrt{u_{t-1,i}}} - \frac{1}{\sqrt{u_{t,i}}}\right) - \alpha \frac{1}{N} \sum_{i=1}^{N} \frac{g_{t,i}}{\sqrt{u_{t,i}}}.$$

Lemma 1 does not display any momentum term in $\frac{1}{N}\sum_{i=1}^{N}\frac{g_{t,i}}{\sqrt{u_{t,i}}}$. This simplification is convenient since it is directly related to the current gradients instead of the exponential average of past gradients.

*Step 2: Smoothness.* Using smoothness assumption A1 involves the following scalar product term: $\kappa_t := \langle \nabla f(Z_t), \frac{1}{N}\sum_{i=1}^{N}\nabla f_i(x_{t,i})/\sqrt{\overline{U}_t}\rangle$ which can be lower bounded by:

$$\kappa_t \geq \frac{1}{2}\left\|\frac{\nabla f(\overline{X}_t)}{\overline{U}_t^{1/4}}\right\|^2 - \frac{3}{2}\left\|\frac{\nabla f(Z_t) - \nabla f(\overline{X}_t)}{\overline{U}_t^{1/4}}\right\|^2 - \frac{3}{2}\left\|\frac{\frac{1}{N}\sum_{i=1}^{N}\nabla f_i(x_{t,i}) - \nabla f(\overline{X}_t)}{\overline{U}_t^{1/4}}\right\|^2.$$

The above inequality substituted in the smoothness condition $f(Z_{t+1}) \leq f(Z_t) + \langle \nabla f(Z_t), Z_{t+1} - Z_t\rangle + \frac{L}{2}\|Z_{t+1} - Z_t\|^2$ yields:

$$\frac{1}{T}\sum_{t=1}^{T}\mathbb{E}\left[\left\|\frac{\nabla f(\overline{X}_t)}{\overline{U}_t^{1/4}}\right\|^2\right] \leq \frac{2}{T\alpha}\mathbb{E}[\Delta_f] + \frac{L}{T\alpha}\sum_{t=1}^{T}\mathbb{E}\left[\|Z_{t+1} - Z_t\|^2\right] + \frac{2}{T}\frac{\beta_1 D_1}{1-\beta_1} + \frac{2D_2}{T} + \frac{3D_3}{T}, \tag{5}$$

where $\Delta_f := \mathbb{E}[f(Z_1)] - \mathbb{E}[f(Z_{T+1})]$ $D_1, D_2$ and $D_3$ are three terms, defined in the supplementary material, and which can be tightly bounded from above. We first bound $D_3$ using the following quantities of interest:

$$\sum_{t=1}^{T}\|Z_t - \overline{X}_t\|^2 \leq T\left(\frac{\beta_1}{1-\beta_1}\right)^2\alpha^2 d\frac{G_\infty^2}{\epsilon} \quad \text{and} \quad \sum_{t=1}^{T}\frac{1}{N}\sum_{i=1}^{N}\|x_{t,i} - \overline{X}_t\|^2 \leq T\alpha^2\left(\frac{1}{1-\lambda}\right)^2 dG_\infty^2\frac{1}{\epsilon}.$$

where $\lambda = \max(|\lambda_2|, |\lambda_N|)$ and recall that $\lambda_i$ is $i$th largest eigenvalue of $W$.

Then, concerning the term $D_2$, few derivations, not detailed here for simplicity, yields:

$$D_2 \leq \frac{G_\infty^2}{N}\mathbb{E}\left[\sum_{t=1}^{T}\frac{1}{2\epsilon^{1.5}}\| - \sum_{l=2}^{N}\tilde{U}_t q_l q_l^T\|_{abs}\right],$$

where $q_l$ is the eigenvector corresponding to $l$th largest eigenvalue of $W$ and $\|\cdot\|_{abs}$ is the entry-wise $L_1$ norm of matrices. We can also show that

$$\sum_{t=1}^{T}\| - \sum_{l=2}^{N}\tilde{U}_t q_l q_l^T\|_{abs} \leq \sqrt{N}\sum_{o=0}^{T-1}\frac{\lambda}{1-\lambda}\|(-\hat{V}_{o-1} + \hat{V}_o)\|_{abs},$$

resulting in an upper bound for $D_2$ proportional to $\sum_{o=0}^{T-1} \|(-\hat{V}_{o-1} + \hat{V}_o)\|_{abs}$. Similarly:

$$D_1 \le G_\infty^2 \frac{1}{2\epsilon^{1.5}} \frac{1}{\sqrt{N}} \mathbb{E}\left[\frac{1}{1-\lambda} \sum_{t=1}^T \|(-\hat{V}_{t-2} + \hat{V}_{t-1})\|_{abs}\right] .$$

*Step 3: Bounding the drift term variance.* An important term that needs upper bounding in our proof is the variance of the gradients multiplied (element-wise) by the adaptive learning rate:

$$\mathbb{E}\left[\left\|\frac{1}{N}\sum_{i=1}^N \frac{g_{t,i}}{\sqrt{u_{t,i}}}\right\|^2\right] \le \mathbb{E}[\|\Gamma_u^f\|^2] + \frac{d}{N}\frac{\sigma^2}{\epsilon} ,$$

where $\Gamma_u^f := 1/N \sum_{i=1}^N \nabla f_i(x_{t,i})/\sqrt{u_{t,i}}$. Two consecutive and simple bounding of the above yields:

$$\sum_{t=1}^T \mathbb{E}[\|\Gamma_u^f\|^2] \le 2\sum_{t=1}^T \mathbb{E}[\|\Gamma_{\overline{U}}^f\|^2] + 2\sum_{t=1}^T \mathbb{E}\left[\frac{1}{N}\sum_{i=1}^N G_\infty^2 \frac{1}{\sqrt{\epsilon}} \left\|\frac{1}{\sqrt{u_{t,i}}} - \frac{1}{\sqrt{\overline{U}_t}}\right\|_1\right]$$

and

$$\sum_{t=1}^T \mathbb{E}[\|\Gamma_{\overline{U}}^f\|^2] \le 2\sum_{t=1}^T \mathbb{E}\left[\left\|\frac{\nabla f(\overline{X}_t)}{\sqrt{\overline{U}_t}}\right\|^2\right] + 2\sum_{t=1}^T \mathbb{E}\left[\left\|\frac{1}{N}\sum_{i=1}^N \frac{\nabla f_i(\overline{X}_t) - \nabla f_i(x_{t,i})}{\sqrt{\overline{U}_t}}\right\|^2\right] . \quad (6)$$

Then, by plugging the LHS of (6) in (5), and further bounding as operated for $D_2, D_3$ (see supplement), we obtain the desired bound in Theorem 2.

**Proof of Theorem 3:** Recall the bound in (3) of Theorem 2. Since Algorithm 3 is a special case of Algorithm 2, the remaining of the proof consists in characterizing the growth rate of $\mathbb{E}[\sum_{t=1}^T \|(-\hat{V}_{t-2} + \hat{V}_{t-1})\|_{abs}]$. By construction, $\hat{V}_t$ is non decreasing, then it can be shown that $\mathbb{E}[\sum_{t=1}^T \|(-\hat{V}_{t-2} + \hat{V}_{t-1})\|_{abs}] = \mathbb{E}[\sum_{i=1}^N \sum_{j=1}^d (-[\hat{v}_{0,i}]_j + [\hat{v}_{T-1,i}]_j)]$. Besides, since for all $t, i$, $\|g_{t,i}\|_\infty \le G_\infty$ and $v_{t,i}$ is an exponential moving average of $g_{k,i}^2, k = 1, 2, \cdots, t$, we have $|[v_{t,i}]_j| \le G_\infty^2$ for all $t, i, j$. By construction of $\hat{V}_t$, we also observe that each element of $\hat{V}_t$ cannot be greater than $G_\infty^2$, i.e. $|[\hat{v}_{t,i}]_j| \le G_\infty^2$ for all $t, i, j$. Given that $[\hat{v}_{0,i}]_j \ge 0$, we have

$$\mathbb{E}\left[\sum_{t=1}^T \|(-\hat{V}_{t-2} + \hat{V}_{t-1})\|_{abs}\right] = \mathbb{E}\left[\sum_{i=1}^N \sum_{j=1}^d (-[\hat{v}_{0,i}]_j + [\hat{v}_{T-1,i}]_j)\right] \le \sum_{i=1}^N \sum_{j=1}^d \mathbb{E}[G_\infty^2] = NdG_\infty^2 .$$

Substituting into (3) yields the desired convergence bound for Algorithm 3.

### 3.4 ILLUSTRATIVE NUMERICAL EXPERIMENTS

In this section, we conduct some experiments to test the performance of Decentralized AMSGrad, developed in Algorithm 3, on both *homogeneous* data and *heterogeneous* data distribution (i.e. the data generating distribution on different nodes are assumed to be different). Comparison with DADAM and the decentralized stochastic gradient descent (DGD) developed in Lian et al. (2017) are conducted. We train a Convolutional Neural Network (CNN) with 3 convolution layers followed by a fully connected layer on MNIST (LeCun, 1998). We set $\epsilon = 10^{-6}$ for both Decentralized AMSGrad and DADAM. The learning rate is chosen from the grid $[10^{-1}, 10^{-2}, 10^{-3}, 10^{-4}, 10^{-5}, 10^{-6}]$ based on validation accuracy for all algorithms. In the following experiments, the graph contains 5 nodes and each node can only communicate with its two adjacent neighbors forming a cycle. Regarding the mixing matrix $W$, we set $W_{ij} = 1/3$ if nodes $i$ and $j$ are neighbors and $W_{ij} = 0$ otherwise. More details and experiments can be found in the supplementary material of our paper.

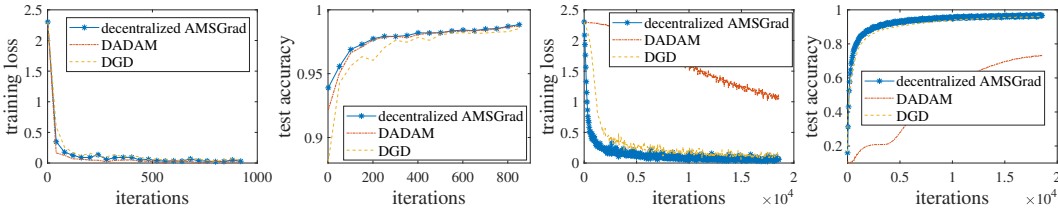

(a) Homogeneous data  (b) Heterogeneous data
Figure 1: Training loss and Testing accuracy for homogeneous and heterogeneous data

*Homogeneous data:* The whole dataset is shuffled and evenly split into different nodes. Such a setting is possible when the nodes are in a computer cluster. We see, Figure 1(a), that decentralized AMSGrad and DADAM perform quite similarly while DGD is much slower both in terms of training loss and test accuracy. Though the (possible) non convergence of DADAM, mentioned in this paper, its performance are empirically good on homogeneous data. The reason is that the adaptive learning rates tend to be similar on different nodes in presence of homogeneous data distribution. We thus compare these algorithms under the heterogeneous regime.

*Heterogeneous data:* Here, each node only contains training data with two labels out of ten. Such a setting is common when data shuffling is prohibited, such as in federated learning. We can see that each algorithm converges significantly slower than with homogeneous data. Especially, the performance of DADAM deteriorates significantly. Decentralized AMSGrad achieves the best training and testing performance in that setting as observed Figure 1(b).

## 4 Extension to AdaGrad

In this section, we provide a decentralized version of AdaGrad (optionally with momentum) converted by Algorithm 2, further supporting the usefulness of the decentralization framework.

The required modification for decentralize AdaGrad is to specify line 4 of Algorithm 2 as

$$\hat{v}_{t,i} = \frac{t-1}{t}\hat{v}_{t-1,i} + \frac{1}{t}g_{t,i}^2$$

which is equivalent to $\hat{v}_{t,i} = \frac{1}{t}\sum_{k=1}^{t}g_{k,i}^2$. Throughout this section, we will call this algorithm decentralized AdaGrad. There are two details in the algorithm worth mentioning. One is that the framework uses momentum $m_{t,i}$ in updates, while original AdaGrad does not use momentum. The momentum can be turned off by setting $\beta_1 = 0$ and the convergence results will hold. The other one is that in decentralized AdaGrad, we use average instead of sum in $\hat{v}_{t,i}$. I.e. $\hat{v}_{t,i} = \frac{1}{t}\sum_{k=1}^{t}g_{k,i}^2$. This is different from original AdaGrad which should use $\hat{v}_{t,i} = \sum_{k=1}^{t}g_{k,i}^2$. The reason is in original AdaGrad, a constant stepsize ($\alpha$ independent of $t$ or $T$) is used with $\hat{v}_{t,i} = \sum_{k=1}^{t}g_{k,i}^2$ and this is equivalent to using a well-known diminishing stepsize sequence $\alpha_t = \frac{1}{\sqrt{t}}$ with $\hat{v}_{t,i} = \frac{1}{t}\sum_{k=1}^{t}g_{k,i}^2$. In our convergence analysis which will be presented later, we will use a constant stepsize $\alpha = O(\frac{1}{\sqrt{T}})$ to replace the diminishing stepsize sequence $\alpha_t = O(\frac{1}{\sqrt{t}})$. Such a replacement is popularly used in SGD analysis to simplify analysis and achieving better convergence rate. In addition, it is easy to modify our theoretical framework to apply diminishing stepsize sequences such as $\alpha_t = O(\frac{1}{\sqrt{t}})$.

The convergence analysis for decentralized AdaGrad is shown in Theorem 4.

**Theorem 4.** *Assume A1-A4. Set $\alpha = \sqrt{N}/\sqrt{Td}$. When $\alpha \leq \frac{\epsilon^{0.5}}{16L}$, decentralized AdaGrad yields the following regret bound*

$$\frac{1}{T}\sum_{t=1}^{T}\mathbb{E}\left[\left\|\frac{\nabla f(\overline{X}_t)}{\overline{U}_t^{1/4}}\right\|^2\right] \leq \frac{C_1'\sqrt{d}}{\sqrt{TN}}D_f' + \frac{C_2'}{T} + \frac{C_3'N^{1.5}}{T^{1.5}d^{0.5}} + \frac{\sqrt{N}(1+\log(T))}{T}\left(dC_4' + \frac{\sqrt{d}}{T^{0.5}}C_5'\right),$$

*where $D_f' := \mathbb{E}[f(Z_1)] - \min_z f(z) + \sigma^2$, $C_1' = C_1$, $C_2' = C_2$, $C_3' = C_3$, $C_4' = C_4G_\infty^2$ and $C_5' = C_5G_\infty^2$. $C_1, C_2, C_3, C_4, C_5$ are defined in Theorem 2 independent of d, T and N. In addition, the consensus of variables at different nodes is given by $\frac{1}{N}\sum_{i=1}^{N}\left\|x_{t,i} - \overline{X}_t\right\|^2 \leq \frac{N}{T}\left(\frac{1}{1-\lambda}\right)^2 G_\infty^2 \frac{1}{\epsilon}$.*

## 5 Conclusion

This paper studies the problem of designing adaptive gradient methods for decentralized training. We propose a unifying algorithmic framework that can convert existing adaptive gradient methods to decentralized settings. With rigorous convergence analysis, we show that if the original algorithm satisfies converges under some minor conditions, the converted algorithm obtained using our proposed framework is guaranteed to converge to stationary points of the regret function. By applying our framework to AMSGrad, we propose the first convergent adaptive gradient methods, namely Decentralized AMSGrad. Experiments show that the proposed algorithm achieves better performance than the baselines.

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

## A  PROOF OF AUXILIARY LEMMAS

**Lemma 1.** *For the sequence defined in* (10)*, we have*

$$Z_{t+1} - Z_t = \alpha \frac{\beta_1}{1-\beta_1} \frac{1}{N} \sum_{i=1}^{N} m_{t-1,i} \odot \left( \frac{1}{\sqrt{u_{t-1,i}}} - \frac{1}{\sqrt{u_{t,i}}} \right) - \alpha \frac{1}{N} \sum_{i=1}^{N} \frac{g_{t,i}}{\sqrt{u_{t,i}}} . \tag{7}$$

**Proof:** By update rule of Algorithm 2, we first have

$$\begin{aligned}
\overline{X}_{t+1} &= \frac{1}{N} \sum_{i=1}^{N} x_{t+1,i} \\
&= \frac{1}{N} \sum_{i=1}^{N} \left( x_{t+0.5,i} - \alpha \frac{m_{t,i}}{\sqrt{u_{t,i}}} \right) \\
&= \frac{1}{N} \sum_{i=1}^{N} \left( \sum_{j=1}^{N} W_{ij} x_{t,j} - \alpha \frac{m_{t,i}}{\sqrt{u_{t,i}}} \right) \\
&\overset{(i)}{=} \left( \frac{1}{N} \sum_{j=1}^{N} x_{t,j} \right) - \frac{1}{N} \sum_{i=1}^{N} \alpha \frac{m_{t,i}}{\sqrt{u_{t,i}}} \\
&= \overline{X}_t - \frac{1}{N} \sum_{i=1}^{N} \alpha \frac{m_{t,i}}{\sqrt{u_{t,i}}} ,
\end{aligned}$$

where (i) is due to an interchange of summation and $\sum_{i=1} W_{ij} = 1$. Then, we have

$$\begin{aligned}
Z_{t+1} - Z_t &= \overline{X}_{t+1} - \overline{X}_t + \frac{\beta_1}{1-\beta_1} (\overline{X}_{t+1} - \overline{X}_t) - \frac{\beta_1}{1-\beta_1} (\overline{X}_{t+1} - \overline{X}_t) \\
&= \frac{1}{1-\beta_1} (\overline{X}_{t+1} - \overline{X}_t) - \frac{\beta_1}{1-\beta_1} (\overline{X}_{t+1} - \overline{X}_t) \\
&= \frac{1}{1-\beta_1} \left( -\frac{1}{N} \sum_{i=1}^{N} \alpha \frac{m_{t,i}}{\sqrt{u_{t,i}}} \right) - \frac{\beta_1}{1-\beta_1} \left( -\frac{1}{N} \sum_{i=1}^{N} \alpha \frac{m_{t-1,i}}{\sqrt{u_{t-1,i}}} \right) \\
&= \frac{1}{1-\beta_1} \left( -\frac{1}{N} \sum_{i=1}^{N} \alpha \frac{\beta_1 m_{t-1,i} + (1-\beta_1) g_{t,i}}{\sqrt{u_{t,i}}} \right) - \frac{\beta_1}{1-\beta_1} \left( -\frac{1}{N} \sum_{i=1}^{N} \alpha \frac{m_{t-1,i}}{\sqrt{u_{t-1,i}}} \right) \\
&= \alpha \frac{\beta_1}{1-\beta_1} \frac{1}{N} \sum_{i=1}^{N} m_{t-1,i} \odot \left( \frac{1}{\sqrt{u_{t-1,i}}} - \frac{1}{\sqrt{u_{t,i}}} \right) - \alpha \frac{1}{N} \sum_{i=1}^{N} \frac{g_{t,i}}{\sqrt{u_{t,i}}} ,
\end{aligned}$$

which is the desired result. $\square$

**Lemma A.1.** *Given a set of numbers $a_1, \cdots, a_n$ and denote their mean to be $\bar{a} = \frac{1}{n} \sum_{i=1}^{n} a_i$. Define $b_i(r) \triangleq = \max(a_i, r)$ and $\bar{b}(r) = \frac{1}{n} \sum_{i=1}^{n} b_i(r)$. For any $r$ and $r'$ with $r' \geq r$ we have*

$$\sum_{i=1}^{n} |b_i(r) - \bar{b}(r)| \geq \sum_{i=1}^{n} |b_i(r') - \bar{b}(r')| \tag{8}$$

*and when $r \leq \min_{i \in [n]} a_i$, we have*

$$\sum_{i=1}^{n} |b_i(r) - \bar{b}(r)| = \sum_{i=1}^{n} |a_i - \bar{a}| . \tag{9}$$

**Proof**: Without loss of generality, assume $a_i \leq a_j$ when $i < j$, i.e. $a_i$ is a non-decreasing sequence. Define

$$h(r) = \sum_{i=1}^{n} |b_i(r) - \bar{b}(r)| = \sum_{i=1}^{n} |\max(a_i, r) - \frac{1}{n} \sum_{j=1}^{n} \max(a_j, r)| .$$

We need to prove that $h$ is a non-increasing function of $r$. First, it is easy to see that $h$ is a continuous function of $r$ with non-differentiable points $r = a_i, i \in [n]$, thus $h$ is a piece-wise linear function.

Next, we will prove that $h(r)$ is non-increasing in each piece. Define $l(r)$ to be the largest index with $a(l(r)) < r$, and $s(r)$ to be the largest index with $a_{s(r)} < \bar{b}(r)$. Note that we have for $i \leq l(r)$, $b_i(r) = r$ and for $i \leq s(r)$ $b_i(r) - \bar{b}(r) \leq 0$ since $a_i$ is a non-decreasing sequence. Therefore, we have

$$h(r) = \sum_{i=1}^{l(r)}(\bar{b}(r) - r) + \sum_{i=l(r)+1}^{s(r)}(\bar{b}(r) - a_i) + \sum_{i=s(r)+1}^{n}(a_i - \bar{b}(r))$$

and

$$\bar{b}(r) = \frac{1}{n}\left(l(r)r + \sum_{i=l(r)+1}^{n} a_i\right).$$

Taking derivative of the above form, we know the derivative of $h(r)$ at differentiable points is

$$h'(r) = l(r)(\frac{l(r)}{n} - 1) + (s(r) - l(r))\frac{l(r)}{n} - (n - s(r))\frac{l(r)}{n}$$
$$= \frac{l(r)}{n}((l(r) - n) + (s(r) - l(r)) - (n - s(r))).$$

Since we have $s(r) \leq n$ we know $(l(r) - n) + (s(r) - l(r)) - (n - s(r)) \leq 0$ and thus

$$h'(r) \leq 0,$$

which means $h(r)$ is non-increasing in each piece. Combining with the fact that $h(r)$ is continuous, (8) is proven. When $r \leq a(i)$, we have $b(i) = \max(a_i, r) = r$, for all $r \in [n]$ and $\bar{b}(r) = \frac{1}{n}\sum_{i=1}^{n} a_i = \bar{a}$ which proves (9). □

## B  PROOF OF THEOREM 2

To prove convergence of the algorithm, we first define an auxiliary sequence

$$Z_t = \overline{X}_t + \frac{\beta_1}{1 - \beta_1}(\overline{X}_t - \overline{X}_{t-1}),\tag{10}$$

with $\overline{X}_0 \triangleq \overline{X}_1$. Since $\mathbb{E}[g_{t,i}] = \nabla f(x_{t,i})$ and $u_{t,i}$ is a function of $G_{1:t-1}$ (which denotes $G_1, G_2, \cdots, G_{t-1}$), we have

$$\mathbb{E}_{G_t|G_{1:t-1}}\left[\frac{1}{N}\sum_{i=1}^{N}\frac{g_{t,i}}{\sqrt{u_{t,i}}}\right] = \frac{1}{N}\sum_{i=1}^{N}\frac{\nabla f_i(x_{t,i})}{\sqrt{u_{t,i}}}.$$

Assuming smoothness (A1) we have

$$f(Z_{t+1}) \leq f(Z_t) + \langle\nabla f(Z_t), Z_{t+1} - Z_t\rangle + \frac{L}{2}\|Z_{t+1} - Z_t\|^2.$$

Using Lemma 1 into the above inequality and take expectation over $G_t$ given $G_{1:t-1}$, we have

$$\mathbb{E}_{G_t|G_{1:t-1}}[f(Z_{t+1})]$$
$$\leq f(Z_t) - \alpha\left\langle\nabla f(Z_t), \frac{1}{N}\sum_{i=1}^{N}\frac{\nabla f_i(x_{t,i})}{\sqrt{u_{t,i}}}\right\rangle + \frac{L}{2}\mathbb{E}_{G_t|G_{1:t-1}}\left[\|Z_{t+1} - Z_t\|^2\right]$$
$$+ \alpha\frac{\beta_1}{1 - \beta_1}\mathbb{E}_{G_t|G_{1:t-1}}\left[\left\langle\nabla f(Z_t), \frac{1}{N}\sum_{i=1}^{N}m_{t-1,i}\odot(\frac{1}{\sqrt{u_{t-1,i}}} - \frac{1}{\sqrt{u_{t,i}}})\right\rangle\right].$$

Then take expectation over $G_{1:t-1}$ and rearrange, we have

$$\alpha \mathbb{E}\left[\left\langle \nabla f(Z_t), \frac{1}{N}\sum_{i=1}^{N}\frac{\nabla f_i(x_{t,i})}{\sqrt{u_{t,i}}}\right\rangle\right] \tag{11}$$

$$\leq \mathbb{E}[f(Z_t)] - \mathbb{E}[f(Z_{t+1})] + \frac{L}{2}\mathbb{E}\left[\|Z_{t+1} - Z_t\|^2\right]$$

$$+ \alpha\frac{\beta_1}{1-\beta_1}\mathbb{E}\left[\left\langle \nabla f(Z_t), \frac{1}{N}\sum_{i=1}^{N}m_{t-1,i}\odot(\frac{1}{\sqrt{u_{t-1,i}}} - \frac{1}{\sqrt{u_{t,i}}})\right\rangle\right]. \tag{12}$$

In addition, we have

$$\left\langle \nabla f(Z_t), \frac{1}{N}\sum_{i=1}^{N}\frac{\nabla f_i(x_{t,i})}{\sqrt{u_{t,i}}}\right\rangle$$

$$= \left\langle \nabla f(Z_t), \frac{1}{N}\sum_{i=1}^{N}\frac{\nabla f_i(x_{t,i})}{\sqrt{\overline{U}_t}}\right\rangle + \left\langle \nabla f(Z_t), \frac{1}{N}\sum_{i=1}^{N}\nabla f_i(x_{t,i})\odot\left(\frac{1}{\sqrt{u_{t,i}}} - \frac{1}{\sqrt{\overline{U}_t}}\right)\right\rangle \tag{13}$$

and the first term on RHS of the equality can be lower bounded as

$$\left\langle \nabla f(Z_t), \frac{1}{N}\sum_{i=1}^{N}\frac{\nabla f_i(x_{t,i})}{\sqrt{\overline{U}_t}}\right\rangle$$

$$= \frac{1}{2}\left\|\frac{\nabla f(Z_t)}{\overline{U}_t^{1/4}}\right\|^2 + \frac{1}{2}\left\|\frac{\frac{1}{N}\sum_{i=1}^{N}\nabla f_i(x_{t,i})}{\overline{U}_t^{1/4}}\right\|^2 - \frac{1}{2}\left\|\frac{\nabla f(Z_t) - \frac{1}{N}\sum_{i=1}^{N}\nabla f_i(x_{t,i})}{\overline{U}_t^{1/4}}\right\|^2$$

$$\geq \frac{1}{4}\left\|\frac{\nabla f(\overline{X}_t)}{\overline{U}_t^{1/4}}\right\|^2 + \frac{1}{4}\left\|\frac{\nabla f(\overline{X}_t)}{\overline{U}_t^{1/4}}\right\|^2 - \frac{1}{2}\left\|\frac{\nabla f(Z_t) - \frac{1}{N}\sum_{i=1}^{N}\nabla f_i(x_{t,i})}{\overline{U}_t^{1/4}}\right\|^2$$

$$- \frac{1}{2}\left\|\frac{\nabla f(Z_t) - \nabla f(\overline{X}_t)}{\overline{U}_t^{1/4}}\right\|^2 - \frac{1}{2}\left\|\frac{\frac{1}{N}\sum_{i=1}^{N}\nabla f_i(x_{t,i}) - \nabla f(\overline{X}_t)}{\overline{U}_t^{1/4}}\right\|^2$$

$$\geq \frac{1}{2}\left\|\frac{\nabla f(\overline{X}_t)}{\overline{U}_t^{1/4}}\right\|^2 - \frac{3}{2}\left\|\frac{\nabla f(Z_t) - \nabla f(\overline{X}_t)}{\overline{U}_t^{1/4}}\right\|^2 - \frac{3}{2}\left\|\frac{\frac{1}{N}\sum_{i=1}^{N}\nabla f_i(x_{t,i}) - \nabla f(\overline{X}_t)}{\overline{U}_t^{1/4}}\right\|^2, \tag{14}$$

where the inequalities are all due to Cauchy-Schwartz. Substituting (14) and (13) into (11), we get

$$\frac{1}{2}\alpha\mathbb{E}\left[\left\|\frac{\nabla f(\overline{X}_t)}{\overline{U}_t^{1/4}}\right\|^2\right] \leq \mathbb{E}[f(Z_t)] - \mathbb{E}[f(Z_{t+1})] + \frac{L}{2}\mathbb{E}\left[\|Z_{t+1} - Z_t\|^2\right]$$

$$+ \alpha\frac{\beta_1}{1-\beta_1}\mathbb{E}\left[\left\langle \nabla f(Z_t), \frac{1}{N}\sum_{i=1}^{N}m_{t-1,i}\odot(\frac{1}{\sqrt{u_{t-1,i}}} - \frac{1}{\sqrt{u_{t,i}}})\right\rangle\right]$$

$$- \alpha\mathbb{E}\left[\left\langle \nabla f(Z_t), \frac{1}{N}\sum_{i=1}^{N}\nabla f_i(x_{t,i})\odot\left(\frac{1}{\sqrt{u_{t,i}}} - \frac{1}{\sqrt{\overline{U}_t}}\right)\right\rangle\right]$$

$$+ \frac{3}{2}\alpha\mathbb{E}\left[\left\|\frac{\frac{1}{N}\sum_{i=1}^{N}\nabla f_i(x_{t,i}) - \nabla f(\overline{X}_t)}{\overline{U}_t^{1/4}}\right\|^2 + \left\|\frac{\nabla f(Z_t) - \nabla f(\overline{X}_t)}{\overline{U}_t^{1/4}}\right\|^2\right].$$

Then sum over the above inequality from $t = 1$ to $T$ and divide both sides by $T\alpha/2$, we have

$$\frac{1}{T}\sum_{t=1}^{T}\mathbb{E}\left[\left\|\frac{\nabla f(\overline{X}_t)}{\overline{U}_t^{1/4}}\right\|^2\right]$$

$$\leq \frac{2}{T\alpha}(\mathbb{E}[f(Z_1)] - \mathbb{E}[f(Z_{T+1})]) + \frac{L}{T\alpha}\sum_{t=1}^{T}\mathbb{E}\left[\|Z_{t+1} - Z_t\|^2\right]$$

$$+ \underbrace{\frac{2}{T}\frac{\beta_1}{1-\beta_1}\sum_{t=1}^{T}\mathbb{E}\left[\left\langle \nabla f(Z_t), \frac{1}{N}\sum_{i=1}^{N}m_{t-1,i}\odot(\frac{1}{\sqrt{u_{t-1,i}}} - \frac{1}{\sqrt{u_{t,i}}})\right\rangle\right]}_{D_1}$$  (15)

$$+ \underbrace{\frac{2}{T}\sum_{t=1}^{T}\mathbb{E}\left[\left\langle \nabla f(Z_t), \frac{1}{N}\sum_{i=1}^{N}\nabla f_i(x_{t,i})\odot\left(\frac{1}{\sqrt{\overline{U}_t}} - \frac{1}{\sqrt{u_{t,i}}}\right)\right\rangle\right]}_{D_2}$$

$$+ \underbrace{\frac{3}{T}\sum_{t=1}^{T}\mathbb{E}\left[\left\|\frac{\frac{1}{N}\sum_{i=1}^{N}\nabla f_i(x_{t,i}) - \nabla f(\overline{X}_t)}{\overline{U}_t^{1/4}}\right\|^2 + \left\|\frac{\nabla f(Z_t) - \nabla f(\overline{X}_t)}{\overline{U}_t^{1/4}}\right\|^2\right]}_{D_3}.$$

Now we need to upper bound all the terms on RHS of the above inequality to get the convergence rate. For the terms composing $D_3$ in (15), we can upper bound them by

$$\left\|\frac{\nabla f(Z_t) - \nabla f(\overline{X}_t)}{\overline{U}_t^{1/4}}\right\|^2 \leq \frac{1}{\min_{j\in[d]}[\overline{U}_t^{1/2}]_j}\left\|\nabla f(Z_t) - \nabla f(\overline{X}_t)\right\|^2$$

$$\leq L\frac{1}{\min_{j\in[d]}[\overline{U}_t^{1/2}]_j}\underbrace{\left\|Z_t - \overline{X}_t\right\|^2}_{D_4}$$  (16)

and

$$\left\|\frac{\frac{1}{N}\sum_{i=1}^{N}\nabla f_i(x_{t,i}) - \nabla f(\overline{X}_t)}{\overline{U}_t^{1/4}}\right\|^2 \leq \frac{1}{\min_{j\in[d]}[\overline{U}_t^{1/2}]_j}\frac{1}{N}\sum_{i=1}^{N}\left\|\nabla f_i(x_{t,i}) - \nabla f(\overline{X}_t)\right\|^2$$

$$\leq L\frac{1}{\min_{j\in[d]}[\overline{U}_t^{1/2}]_j}\frac{1}{N}\underbrace{\sum_{i=1}^{N}\left\|x_{t,i} - \overline{X}_t\right\|^2}_{D_5},$$  (17)

using Jensen's inequality, Lipschitz continuity of $f_i$, and the fact that $f = \frac{1}{N}\sum_{i=1}^{N}f_i$. Next we need to bound $D_4$ and $D_5$. Recall the update rule of $X_t$, we have

$$X_t = X_{t-1}W - \alpha\frac{M_{t-1}}{\sqrt{U_{t-1}}} = X_1W^{t-1} - \alpha\sum_{k=0}^{t-2}\frac{M_{t-k-1}}{\sqrt{U_{t-k-1}}}W^k,$$  (18)

where we define $W^0 = \mathbf{I}$. Since $W$ is a symmetric matrix, we can decompose it as $W = Q\Lambda Q^T$ where $Q$ is a orthonormal matrix and $\Lambda$ is a diagonal matrix whose diagonal elements correspond to eigenvalues of $W$ in an descending order, i.e. $\Lambda_{ii} = \lambda_i$ with $\lambda_i$ being $i$th largest eigenvalue of $W$. In addition, because $W$ is a doubly stochastic matrix, we know $\lambda_1 = 1$ and $q_1 = \frac{\mathbf{1}_N}{\sqrt{N}}$. With eigen-decomposition of $W$, we can rewrite $D_5$ as

$$\sum_{i=1}^{N}\left\|x_{t,i} - \overline{X}_t\right\|^2 = \|X_t - \overline{X}_t\mathbf{1}_N^T\|_F^2 = \|X_tQQ^T - X_t\frac{1}{N}\mathbf{1}_N\mathbf{1}_N^T\|_F^2 = \sum_{l=2}^{N}\|X_tq_l\|^2.$$  (19)

In addition, we can rewrite (18) as

$$X_t = X_1 W^{t-1} - \alpha \sum_{k=0}^{t-2} \frac{M_{t-k-1}}{\sqrt{U_{t-k-1}}} W^k = X_1 - \alpha \sum_{k=0}^{t-2} \frac{M_{t-k-1}}{\sqrt{U_{t-k-1}}} Q \Lambda^k Q^T \,, \tag{20}$$

where the last equality is because $x_{1,i} = x_{1,j}$, for all $i, j$ and thus $X_1 W = X_1$. Then we have when $l > 1$,

$$X_t q_l = (X_1 - \alpha \sum_{k=0}^{t-2} \frac{M_{t-k-1}}{\sqrt{U_{t-k-1}}} Q \Lambda^k Q^T) q_l = -\alpha \sum_{k=0}^{t-2} \frac{M_{t-k-1}}{\sqrt{U_{t-k-1}}} q_l \lambda_l^k \,, \tag{21}$$

since $Q$ is orthonormal and $X_1 q_l = x_{1,1} \mathbf{1}_N^T q_l = x_{1,1} \sqrt{N} q_1^T q_l = 0$, for all $l \neq 1$ .

Combining (19) and (21), we have

$$D_5 = \sum_{i=1}^N \left\| x_{t,i} - \overline{X}_t \right\|^2 = \sum_{l=2}^N \| X_t q_l \|^2$$

$$= \sum_{l=2}^N \alpha^2 \left\| \sum_{k=0}^{t-2} \frac{M_{t-k-1}}{\sqrt{U_{t-k-1}}} \lambda_l^k q_l \right\|^2 \tag{22}$$

$$\leq \alpha^2 \left( \frac{1}{1-\lambda} \right)^2 N d G_\infty^2 \frac{1}{\epsilon} \,,$$

where the last inequality follows from the fact that $g_{t,i} \leq G_\infty$, $\|q_l\| = 1$, and $|\lambda_l| \leq \lambda < 1$. Now let us turn to $D_4$, it can be rewritten as

$$\left\| Z_t - \overline{X}_t \right\|^2 = \left\| \frac{\beta_1}{1-\beta_1} (\overline{X}_t - \overline{X}_{t-1}) \right\|^2 = \left( \frac{\beta_1}{1-\beta_1} \right)^2 \alpha^2 \left\| \frac{1}{N} \sum_{i=1}^N \frac{m_{t-1,i}}{\sqrt{u_{t-1,i}}} \right\|^2$$

$$\leq \left( \frac{\beta_1}{1-\beta_1} \right)^2 \alpha^2 d \frac{G_\infty^2}{\epsilon} \,. \tag{23}$$

Now we know both $D_4$ and $D_5$ are in the order of $\mathcal{O}(\alpha^2)$ and thus $D_3$ is in the order of $\mathcal{O}(\alpha^2)$. Next we will bound $D_2$ and $D_1$. Define $G_1 \triangleq \max_{t \in [T]} \max_{i \in [N]} \|\nabla f_i(x_{t,i})\|_\infty$, $G_2 \triangleq \max_{t \in [T]} \|\nabla f(Z_t)\|_\infty$, $G_3 \triangleq \max_{t \in [T]} \max_{i \in [N]} \|g_{t,i}\|_\infty$ and $G_\infty = \max(G_1, G_2, G_3)$. Then we have

$$D_2 = \sum_{t=1}^T \mathbb{E} \left[ \left\langle \nabla f(Z_t), \frac{1}{N} \sum_{i=1}^N \nabla f_i(x_{t,i}) \odot \left( \frac{1}{\sqrt{\overline{U}_t}} - \frac{1}{\sqrt{u_{t,i}}} \right) \right\rangle \right]$$

$$\leq \sum_{t=1}^T \mathbb{E} \left[ G_\infty^2 \frac{1}{N} \sum_{i=1}^N \sum_{j=1}^d \left| \frac{1}{\sqrt{[\overline{U}_t]_j}} - \frac{1}{\sqrt{[u_{t,i}]_j}} \right| \right]$$

$$= \sum_{t=1}^T \mathbb{E} \left[ G_\infty^2 \frac{1}{N} \sum_{i=1}^N \sum_{j=1}^d \left| \frac{1}{\sqrt{[\overline{U}_t]_j}} - \frac{1}{\sqrt{[u_{t,i}]_j}} \right| \frac{\sqrt{[\overline{U}_t]_j} + \sqrt{[u_{t,i}]_j}}{\sqrt{[\overline{U}_t]_j} + \sqrt{[u_{t,i}]_j}} \right] \tag{24}$$

$$= \sum_{t=1}^T \mathbb{E} \left[ G_\infty^2 \frac{1}{N} \sum_{i=1}^N \sum_{j=1}^d \left| \frac{[\overline{U}_t]_j - [u_{t,i}]_j}{[\overline{U}_t]_j \sqrt{[u_{t,i}]_j} + \sqrt{[\overline{U}_t]_j [u_{t,i}]_j}} \right| \right]$$

$$\leq \mathbb{E} \bigg[ \underbrace{\sum_{t=1}^T G_\infty^2 \frac{1}{N} \sum_{i=1}^N \sum_{j=1}^d \left| \frac{[\overline{U}_t]_j - [u_{t,i}]_j}{2\epsilon^{1.5}} \right|}_{D_6} \bigg] \,,$$

where the last inequality is due to $[u_{t,i}]_j \geq \epsilon$, for all $t, i, j$. To simplify notations, define $\|A\|_{abs} = \sum_{i,j} |A_{ij}|$ to be the entry-wise $L_1$ norm of a matrix $A$, then we obtain

$$
\begin{aligned}
D_6 &\leq \frac{G_\infty^2}{N} \sum_{t=1}^{T} \frac{1}{2\epsilon^{1.5}} \|\overline{U}_t \mathbf{1}^T - U_t\|_{abs} \leq \frac{G_\infty^2}{N} \sum_{t=1}^{T} \frac{1}{2\epsilon^{1.5}} \|\overline{\tilde{U}}_t \mathbf{1}^T - \tilde{U}_t\|_{abs} \\
&= \frac{G_\infty^2}{N} \sum_{t=1}^{T} \frac{1}{2\epsilon^{1.5}} \|\tilde{U}_t \frac{1}{N} \mathbf{1}_N \mathbf{1}_N^T - \tilde{U}_t Q Q^T\|_{abs} \\
&= \frac{G_\infty^2}{N} \sum_{t=1}^{T} \frac{1}{2\epsilon^{1.5}} \| - \sum_{l=2}^{N} \tilde{U}_t q_l q_l^T\|_{abs} ,
\end{aligned}
$$

where the second inequality is due to Lemma A.1, introduced Section A, and the fact that $U_t = \max(\tilde{U}_t, \epsilon)$ (element-wise max operator). Recall from update rule of $U_t$, by defining $\hat{V}_{-1} \triangleq \hat{V}_0$ and $U_0 \triangleq U_{1/2}$, we have for all $t \geq 0$, $\tilde{U}_{t+1} = (\tilde{U}_t - \hat{V}_{t-1} + \hat{V}_t)W$. Thus, we obtain

$$
\tilde{U}_t = \tilde{U}_0 W^t + \sum_{k=1}^{t} (-\hat{V}_{t-1-k} + \hat{V}_{t-k})W^k = \tilde{U}_0 + \sum_{k=1}^{t} (-\hat{V}_{t-1-k} + \hat{V}_{t-k})Q\Lambda^k Q^T .
$$

Then we further obtain when $l \neq 1$,

$$
\tilde{U}_t q_l = (\tilde{U}_0 + \sum_{k=1}^{t} (-\hat{V}_{t-1-k} + \hat{V}_{t-k})Q\Lambda^k Q^T)q_l = \sum_{k=1}^{t} (-\hat{V}_{t-1-k} + \hat{V}_{t-k})q_l \lambda_l^k ,
$$

where the last equality is due to the definition $\tilde{U}_0 \triangleq U_{1/2} = \epsilon \mathbf{1_d} \mathbf{1}_N^T = \sqrt{N} \epsilon \mathbf{1_d} \mathbf{1}_N^T$ (recall that $q_1 = \frac{1}{\sqrt{N}} \mathbf{1}_N^T$) and $q_i^T q_j = 0$ when $i \neq j$. Note that by definition of $\|\cdot\|_{abs}$, we have for all $A, B$, $\|A + B\|_{abs} \leq \|A\|_{abs} + \|B\|_{abs}$, then

$$
\begin{aligned}
D_6 &\leq \frac{G_\infty^2}{N} \sum_{t=1}^{T} \frac{1}{2\epsilon^{1.5}} \| - \sum_{l=2}^{N} \tilde{U}_t q_l q_l^T\|_{abs} \\
&= \frac{G_\infty^2}{N} \sum_{t=1}^{T} \frac{1}{2\epsilon^{1.5}} \| - \sum_{k=1}^{t} (-\hat{V}_{t-1-k} + \hat{V}_{t-k}) \sum_{l=2}^{N} q_l \lambda_l^k q_l^T\|_{abs} \\
&\leq \frac{G_\infty^2}{N} \sum_{t=1}^{T} \frac{1}{2\epsilon^{1.5}} \sum_{k=1}^{t} \sum_{j=1}^{d} \| \sum_{l=2}^{N} q_l \lambda_l^k q_l^T\|_1 \|(-\hat{V}_{t-1-k} + \hat{V}_{t-k})^T e_j\|_1 \\
&\leq \frac{G_\infty^2}{N} \sum_{t=1}^{T} \frac{1}{2\epsilon^{1.5}} \sum_{k=1}^{t} \sum_{j=1}^{d} \sqrt{N} \| \sum_{l=2}^{N} q_l \lambda_l^k q_l^T\|_2 \|(-\hat{V}_{t-1-k} + \hat{V}_{t-k})^T e_j\|_1 \\
&\leq \frac{G_\infty^2}{N} \sum_{t=1}^{T} \frac{1}{2\epsilon^{1.5}} \sum_{k=1}^{t} \sum_{j=1}^{d} \|(-\hat{V}_{t-1-k} + \hat{V}_{t-k})^T e_j\|_1 \sqrt{N} \lambda^k \\
&= \frac{G_\infty^2}{N} \sum_{t=1}^{T} \frac{1}{2\epsilon^{1.5}} \sum_{k=1}^{t} \|(-\hat{V}_{t-1-k} + \hat{V}_{t-k})\|_{abs} \sqrt{N} \lambda^k \\
&= \frac{G_\infty^2}{N} \frac{1}{2\epsilon^{1.5}} \sum_{o=0}^{T-1} \sum_{t=o+1}^{T} \|(-\hat{V}_{o-1} + \hat{V}_o)\|_{abs} \sqrt{N} \lambda^{t-o} \\
&\leq \frac{G_\infty^2}{\sqrt{N}} \frac{1}{2\epsilon^{1.5}} \sum_{o=0}^{T-1} \frac{\lambda}{1-\lambda} \|(-\hat{V}_{o-1} + \hat{V}_o)\|_{abs} ,
\end{aligned}
\tag{25}
$$

where $\lambda = \max(|\lambda_2|, |\lambda_N|)$. Combining (24) and (25), we have

$$
D_2 \leq \frac{G_\infty^2}{\sqrt{N}} \frac{1}{2\epsilon^{1.5}} \frac{\lambda}{1-\lambda} \mathbb{E} \left[ \sum_{o=0}^{T-1} \|(-\hat{V}_{o-1} + \hat{V}_o)\|_{abs} \right] .
$$

Now we need to bound $D_1$, we have

$$
\begin{aligned}
D_1 &= \sum_{t=1}^{T} \mathbb{E}\left[\left\langle \nabla f(Z_t), \frac{1}{N}\sum_{i=1}^{N} m_{t-1,i} \odot \left(\frac{1}{\sqrt{u_{t-1,i}}} - \frac{1}{\sqrt{u_{t,i}}}\right)\right\rangle\right] \\
&\leq \sum_{t=1}^{T} \mathbb{E}\left[G_\infty^2 \frac{1}{N}\sum_{i=1}^{N}\sum_{j=1}^{d}\left|\frac{1}{\sqrt{[u_{t-1,i}]_j}} - \frac{1}{\sqrt{[u_{t,i}]_j}}\right|\right] \\
&= \sum_{t=1}^{T} \mathbb{E}\left[G_\infty^2 \frac{1}{N}\sum_{i=1}^{N}\sum_{j=1}^{d}\left|\left(\frac{1}{\sqrt{[u_{t-1,i}]_j}} - \frac{1}{\sqrt{[u_{t,i}]_j}}\right)\frac{\sqrt{[u_{t,i}]_j} + \sqrt{[u_{t-1,i}]_j}}{\sqrt{[u_{t,i}]_j} + \sqrt{[u_{t-1,i}]_j}}\right|\right] \\
&\leq \sum_{t=1}^{T} \mathbb{E}\left[G_\infty^2 \frac{1}{N}\sum_{i=1}^{N}\sum_{j=1}^{d}\left|\frac{1}{2\epsilon^{1.5}}\left([u_{t-1,i}]_j - [u_{t,i}]_j\right)\right|\right] \\
&\overset{(a)}{\leq} \sum_{t=1}^{T} \mathbb{E}\left[G_\infty^2 \frac{1}{N}\sum_{i=1}^{N}\sum_{j=1}^{d}\frac{1}{2\epsilon^{1.5}}\left|\left([\tilde{u}_{t-1,i}]_j - [\tilde{u}_{t,i}]_j\right)\right|\right] \\
&= G_\infty^2 \frac{1}{2\epsilon^{1.5}}\frac{1}{N}\mathbb{E}\left[\sum_{t=1}^{T}\|\tilde{U}_{t-1} - \tilde{U}_t\|_{abs}\right],
\end{aligned}
\tag{26}
$$

where $(a)$ is due to $[\tilde{u}_{t-1,i}]_j = \max([u_{t-1,i}]_j, \epsilon)$ and the function $\max(\cdot, \epsilon)$ is 1-Lipschitz. In addition, by update rule of $U_t$, we have

$$
\begin{aligned}
&\sum_{t=1}^{T}\|\tilde{U}_{t-1} - \tilde{U}_t\|_{abs} \\
&= \sum_{t=1}^{T}\|\tilde{U}_{t-1} - (\tilde{U}_{t-1} - \hat{V}_{t-2} + \hat{V}_{t-1})W\|_{abs} \\
&= \sum_{t=1}^{T}\|\tilde{U}_{t-1}(QQ^T - Q\Lambda Q^T) + (-\hat{V}_{t-2} + \hat{V}_{t-1})W\|_{abs} \\
&= \sum_{t=1}^{T}\|\tilde{U}_{t-1}(\sum_{l=2}^{N} q_l(1 - \lambda_l)q_l^T) + (-\hat{V}_{t-2} + \hat{V}_{t-1})W\|_{abs} \\
&\leq \sum_{t=1}^{T}\|\sum_{k=1}^{t-1}(-\hat{V}_{t-2-k} + \hat{V}_{t-1-k})\sum_{l=2}^{N} q_l\lambda_l^k(1 - \lambda_l)q_l^T\|_{abs} + \sum_{t=1}^{T}\|(-\hat{V}_{t-2} + \hat{V}_{t-1})W\|_{abs} \\
&\leq \sum_{t=1}^{T}\left(\sum_{k=1}^{t-1}\| - \hat{V}_{t-2-k} + \hat{V}_{t-1-k}\|_{abs}\sqrt{N}\lambda^k\right) + \sum_{t=1}^{T}\|(-\hat{V}_{t-2} + \hat{V}_{t-1})\|_{abs} \\
&= \sum_{t=1}^{T}\left(\sum_{o=1}^{t-1}\| - \hat{V}_{o-2} + \hat{V}_{o-1}\|_{abs}\sqrt{N}\lambda^{t-o}\right) + \sum_{t=1}^{T}\|(-\hat{V}_{t-2} + \hat{V}_{t-1})\|_{abs} \\
&= \sum_{o=1}^{T-1}\sum_{t=o+1}^{T}\left(\| - \hat{V}_{o-2} + \hat{V}_{o-1}\|_{abs}\sqrt{N}\lambda^{t-o}\right) + \sum_{t=1}^{T}\|(-\hat{V}_{t-2} + \hat{V}_{t-1})\|_{abs} \\
&\leq \sum_{o=1}^{T-1}\frac{\lambda}{1-\lambda}\left(\| - \hat{V}_{o-2} + \hat{V}_{o-1}\|_{abs}\sqrt{N}\right) + \sum_{t=1}^{T}\|(-\hat{V}_{t-2} + \hat{V}_{t-1})\|_{abs} \\
&\leq \frac{1}{1-\lambda}\sum_{t=1}^{T}\|(-\hat{V}_{t-2} + \hat{V}_{t-1})\|_{abs}\sqrt{N}.
\end{aligned}
\tag{27}
$$

Combining (26) and (27), we have

$$D_1 \leq G_\infty^2 \frac{1}{2\epsilon^{1.5}} \frac{1}{N} \mathbb{E} \left[ \frac{1}{1-\lambda} \sum_{t=1}^{T} \|(-\hat{V}_{t-2} + \hat{V}_{t-1})\|_{abs} \sqrt{N} \right]. \tag{28}$$

What remains is to bound $\sum_{t=1}^{T} \mathbb{E} \left[ \|Z_{t+1} - Z_t\|^2 \right]$. By update rule of $Z_t$, we have

$$
\begin{aligned}
&\|Z_{t+1} - Z_t\|^2 \\
&= \left\| \alpha \frac{\beta_1}{1-\beta_1} \frac{1}{N} \sum_{i=1}^{N} m_{t-1,i} \odot \left( \frac{1}{\sqrt{u_{t-1,i}}} - \frac{1}{\sqrt{u_{t,i}}} \right) - \alpha \frac{1}{N} \sum_{i=1}^{N} \frac{g_{t,i}}{\sqrt{u_{t,i}}} \right\|^2 \\
&\leq 2\alpha^2 \left\| \frac{\beta_1}{1-\beta_1} \frac{1}{N} \sum_{i=1}^{N} m_{t-1,i} \odot \left( \frac{1}{\sqrt{u_{t-1,i}}} - \frac{1}{\sqrt{u_{t,i}}} \right) \right\|^2 + 2\alpha^2 \left\| \frac{1}{N} \sum_{i=1}^{N} \frac{g_{t,i}}{\sqrt{u_{t,i}}} \right\|^2 \\
&\leq 2\alpha^2 \left( \frac{\beta_1}{1-\beta_1} \right)^2 G_\infty^2 \frac{1}{N} \sum_{i=1}^{N} \sum_{j=1}^{d} \frac{1}{\sqrt{\epsilon}} \left| \frac{1}{\sqrt{[u_{t-1,i}]_j}} - \frac{1}{\sqrt{[u_{t,i}]_j}} \right| + 2\alpha^2 \left\| \frac{1}{N} \sum_{i=1}^{N} \frac{g_{t,i}}{\sqrt{u_{t,i}}} \right\|^2 \\
&\leq 2\alpha^2 \left( \frac{\beta_1}{1-\beta_1} \right)^2 G_\infty^2 \frac{1}{N} \sum_{i=1}^{N} \sum_{j=1}^{d} \frac{1}{\sqrt{\epsilon}} \left| \frac{[u_{t,i}]_j - [u_{t-1,i}]_j}{2\epsilon^{1.5}} \right| + 2\alpha^2 \left\| \frac{1}{N} \sum_{i=1}^{N} \frac{g_{t,i}}{\sqrt{u_{t,i}}} \right\|^2 \\
&\leq 2\alpha^2 \left( \frac{\beta_1}{1-\beta_1} \right)^2 G_\infty^2 \frac{1}{N} \sum_{i=1}^{N} \sum_{j=1}^{d} \frac{1}{2\epsilon^2} |[\tilde{u}_{t,i}]_j - [\tilde{u}_{t-1,i}]_j| + 2\alpha^2 \left\| \frac{1}{N} \sum_{i=1}^{N} \frac{g_{t,i}}{\sqrt{u_{t,i}}} \right\|^2 \\
&= 2\alpha^2 \left( \frac{\beta_1}{1-\beta_1} \right)^2 G_\infty^2 \frac{1}{N} \frac{1}{2\epsilon^2} \|\tilde{U}_t - \tilde{U}_{t-1}\|_{abs} + 2\alpha^2 \left\| \frac{1}{N} \sum_{i=1}^{N} \frac{g_{t,i}}{\sqrt{u_{t,i}}} \right\|^2,
\end{aligned}
\tag{29}
$$

where the last inequality is again due to the definition that $[\tilde{u}_{t,i}]_j = \max([u_{t,i}]_j, \epsilon)$ and the fact that $\max(\cdot, \epsilon)$ is 1-Lipschitz. Then, we have

$$
\begin{aligned}
&\sum_{t=1}^{T} \mathbb{E}[\|Z_{t+1} - Z_t\|^2] \\
&\leq 2\alpha^2 \left( \frac{\beta_1}{1-\beta_1} \right)^2 G_\infty^2 \frac{1}{N} \frac{1}{2\epsilon^2} \mathbb{E} \left[ \sum_{t=1}^{T} \|\tilde{U}_t - \tilde{U}_{t-1}\|_{abs} \right] + 2\alpha^2 \sum_{t=1}^{T} \mathbb{E} \left[ \left\| \frac{1}{N} \sum_{i=1}^{N} \frac{g_{t,i}}{\sqrt{u_{t,i}}} \right\|^2 \right] \\
&\leq \alpha^2 \left( \frac{\beta_1}{1-\beta_1} \right)^2 \frac{G_\infty^2}{\sqrt{N}} \frac{1}{\epsilon^2} \frac{1}{1-\lambda} \mathbb{E} \left[ \sum_{t=1}^{T} \|(-\hat{V}_{t-2} + \hat{V}_{t-1})\|_{abs} \right] + 2\alpha^2 \sum_{t=1}^{T} \mathbb{E} \left[ \left\| \frac{1}{N} \sum_{i=1}^{N} \frac{g_{t,i}}{\sqrt{u_{t,i}}} \right\|^2 \right],
\end{aligned}
$$

where the last inequality is due to (27).

We now bound the last term on RHS of the above inequality. A trivial bound can be

$$\sum_{t=1}^{T} \left\| \frac{1}{N} \sum_{i=1}^{N} \frac{g_{t,i}}{\sqrt{u_{t,i}}} \right\|^2 \leq \sum_{t=1}^{T} dG_\infty^2 \frac{1}{\epsilon},$$

due to $\|g_{t,i}\| \leq G_\infty$ and $[u_{t,i}]_j \geq \epsilon$, for all $j$ (verified from update rule of $u_{t,i}$ and the assumption that $[v_{t,i}]_j \geq \epsilon$, for all $i$). However, the above bound is independent of $N$, to get a better bound, we

need a more involved analysis to show its dependency on $N$. To do this, we first notice that

$$\mathbb{E}_{G_t|G_{1:t-1}}\left[\left\|\frac{1}{N}\sum_{i=1}^{N}\frac{g_{t,i}}{\sqrt{u_{t,i}}}\right\|^2\right]$$

$$=\mathbb{E}_{G_t|G_{1:t-1}}\left[\frac{1}{N^2}\sum_{i=1}^{N}\sum_{j=1}^{N}\left\langle\frac{\nabla f_i(x_{t,i})+\xi_{t,i}}{\sqrt{u_{t,i}}},\frac{\nabla f_j(x_{t,j})+\xi_{t,j}}{\sqrt{u_{t,j}}}\right\rangle\right]$$

$$\overset{(a)}{=}\mathbb{E}_{G_t|G_{1:t-1}}\left[\left\|\frac{1}{N}\sum_{i=1}^{N}\frac{\nabla f_i(x_{t,i})}{\sqrt{u_{t,i}}}\right\|^2\right]+\mathbb{E}_{G_t|G_{1:t-1}}\left[\frac{1}{N^2}\sum_{i=1}^{N}\left\|\frac{\xi_{t,i}}{\sqrt{u_{t,i}}}\right\|^2\right]$$

$$\overset{(b)}{=}\left\|\frac{1}{N}\sum_{i=1}^{N}\frac{\nabla f_i(x_{t,i})}{\sqrt{u_{t,i}}}\right\|^2+\frac{1}{N^2}\sum_{i=1}^{N}\sum_{l=1}^{d}\frac{\mathbb{E}_{G_t|G_{1:t-1}}[[\xi_{t,i}]_l^2]}{[u_{t,i}]_l}$$

$$\overset{(c)}{\leq}\left\|\frac{1}{N}\sum_{i=1}^{N}\frac{\nabla f_i(x_{t,i})}{\sqrt{u_{t,i}}}\right\|^2+\frac{d}{N}\frac{\sigma^2}{\epsilon},$$

where (a) is due to $\mathbb{E}_{G_t|G_{1:t-1}}[\xi_{t,i}]=0$ and $\xi_{t,i}$ is independent of $x_{t,j}$, $u_{t,j}$ for all $j$, and $\xi_j$, for all $j\neq i$, (b) comes from the fact that $x_{t,i}$, $u_{t,i}$ are fixed given $G_{1:t}$, (c) is due to $\mathbb{E}_{G_t|G_{1:t-1}}[[\xi_{t,i}]_l^2]\leq\sigma^2$ and $[u_{t.i}]_l\geq\epsilon$ by definition. Then we have

$$\mathbb{E}\left[\left\|\frac{1}{N}\sum_{i=1}^{N}\frac{g_{t,i}}{\sqrt{u_{t,i}}}\right\|^2\right]=\mathbb{E}_{G_{1:t-1}}\left[\mathbb{E}_{G_t|G_{1:t-1}}\left[\left\|\frac{1}{N}\sum_{i=1}^{N}\frac{g_{t,i}}{\sqrt{u_{t,i}}}\right\|^2\right]\right]$$

$$\leq\mathbb{E}_{G_{1:t-1}}\left[\left\|\frac{1}{N}\sum_{i=1}^{N}\frac{\nabla f_i(x_{t,i})}{\sqrt{u_{t,i}}}\right\|^2+\frac{d}{N}\frac{\sigma^2}{\epsilon}\right]$$

$$=\mathbb{E}\left[\left\|\frac{1}{N}\sum_{i=1}^{N}\frac{\nabla f_i(x_{t,i})}{\sqrt{u_{t,i}}}\right\|^2\right]+\frac{d}{N}\frac{\sigma^2}{\epsilon}. \tag{30}$$

In traditional analysis of SGD-like distributed algorithms, the term corresponding to $\mathbb{E}\left[\left\|\frac{1}{N}\sum_{i=1}^{N}\frac{\nabla f_i(x_{t,i})}{\sqrt{u_{t,i}}}\right\|^2\right]$ will be merged with the first order descent when the stepsize is chosen to be small enough. However, in our case, the term cannot be merged because it is different from the first order descent in our algorithm. A brute-force upper bound is possible but this will lead to a worse convergence rate in terms of $N$. Thus, we need a more detailed analysis for the term in the following.

$$\mathbb{E}\left[\left\|\frac{1}{N}\sum_{i=1}^{N}\frac{\nabla f_i(x_{t,i})}{\sqrt{u_{t,i}}}\right\|^2\right]$$

$$=\mathbb{E}\left[\left\|\frac{1}{N}\sum_{i=1}^{N}\frac{\nabla f_i(x_{t,i})}{\sqrt{\overline{U}_t}}+\frac{1}{N}\sum_{i=1}^{N}\nabla f_i(x_{t,i})\odot\left(\frac{1}{\sqrt{u_{t,i}}}-\frac{1}{\sqrt{\overline{U}_t}}\right)\right\|^2\right]$$

$$\leq 2\mathbb{E}\left[\left\|\frac{1}{N}\sum_{i=1}^{N}\frac{\nabla f_i(x_{t,i})}{\sqrt{\overline{U}_t}}\right\|^2\right]+2\mathbb{E}\left[\left\|\frac{1}{N}\sum_{i=1}^{N}\nabla f_i(x_{t,i})\odot\left(\frac{1}{\sqrt{u_{t,i}}}-\frac{1}{\sqrt{\overline{U}_t}}\right)\right\|^2\right]$$

$$\leq 2\mathbb{E}\left[\left\|\frac{1}{N}\sum_{i=1}^{N}\frac{\nabla f_i(x_{t,i})}{\sqrt{\overline{U}_t}}\right\|^2\right]+2\mathbb{E}\left[\frac{1}{N}\sum_{i=1}^{N}\left\|\nabla f_i(x_{t,i})\odot\left(\frac{1}{\sqrt{u_{t,i}}}-\frac{1}{\sqrt{\overline{U}_t}}\right)\right\|^2\right]$$

$$\leq 2\mathbb{E}\left[\left\|\frac{1}{N}\sum_{i=1}^{N}\frac{\nabla f_i(x_{t,i})}{\sqrt{\overline{U}_t}}\right\|^2\right] + 2\mathbb{E}\left[\frac{1}{N}\sum_{i=1}^{N}G_\infty^2\frac{1}{\sqrt{\epsilon}}\left\|\frac{1}{\sqrt{u_{t,i}}} - \frac{1}{\sqrt{\overline{U}_t}}\right\|_1\right].$$

Summing over $T$, we have

$$\sum_{t=1}^{T}\mathbb{E}\left[\left\|\frac{1}{N}\sum_{i=1}^{N}\frac{\nabla f_i(x_{t,i})}{\sqrt{u_{t,i}}}\right\|^2\right]$$

$$\leq 2\sum_{t=1}^{T}\mathbb{E}\left[\left\|\frac{1}{N}\sum_{i=1}^{N}\frac{\nabla f_i(x_{t,i})}{\sqrt{\overline{U}_t}}\right\|^2\right] + 2\sum_{t=1}^{T}\mathbb{E}\left[\frac{1}{N}\sum_{i=1}^{N}G_\infty^2\frac{1}{\sqrt{\epsilon}}\left\|\frac{1}{\sqrt{u_{t,i}}} - \frac{1}{\sqrt{\overline{U}_t}}\right\|_1\right]. \tag{31}$$

For the last term on RHS of (31), we can bound it similarly as what we did for $D_2$ from (24) to (25), which yields

$$\sum_{t=1}^{T}\mathbb{E}\left[\frac{1}{N}\sum_{i=1}^{N}G_\infty^2\frac{1}{\sqrt{\epsilon}}\left\|\frac{1}{\sqrt{u_{t,i}}} - \frac{1}{\sqrt{\overline{U}_t}}\right\|_1\right] \leq \sum_{t=1}^{T}\mathbb{E}\left[\frac{1}{N}\sum_{i=1}^{N}G_\infty^2\frac{1}{\sqrt{\epsilon}}\frac{1}{2\epsilon^{1.5}}\|u_{t,i} - \overline{U}_t\|_1\right]$$

$$= \sum_{t=1}^{T}\mathbb{E}\left[\frac{1}{N}G_\infty^2\frac{1}{2\epsilon^2}\|\overline{U}_t\mathbf{1}^T - U_t\|_{abs}\right]$$

$$\leq \sum_{t=1}^{T}\mathbb{E}\left[\frac{1}{N}G_\infty^2\frac{1}{2\epsilon^2}\| - \sum_{l=2}^{N}\tilde{U}_t q_l q_l^T\|_{abs}\right]$$

$$\leq \frac{1}{\sqrt{N}}G_\infty^2\frac{1}{2\epsilon^2}\mathbb{E}\left[\sum_{o=0}^{T-1}\frac{\lambda}{1-\lambda}\|(-\hat{V}_{o-1} + \hat{V}_o)\|_{abs}\right]. \tag{32}$$

Further, we have

$$\sum_{t=1}^{T}\mathbb{E}\left[\left\|\frac{1}{N}\sum_{i=1}^{N}\frac{\nabla f_i(x_{t,i})}{\sqrt{\overline{U}_t}}\right\|^2\right]$$

$$\leq 2\sum_{t=1}^{T}\mathbb{E}\left[\left\|\frac{1}{N}\sum_{i=1}^{N}\frac{\nabla f_i(\overline{X}_t)}{\sqrt{\overline{U}_t}}\right\|^2\right] + 2\sum_{t=1}^{T}\mathbb{E}\left[\left\|\frac{1}{N}\sum_{i=1}^{N}\frac{\nabla f_i(\overline{X}_t) - \nabla f_i(x_{t,i})}{\sqrt{\overline{U}_t}}\right\|^2\right]$$

$$= 2\sum_{t=1}^{T}\mathbb{E}\left[\left\|\frac{\nabla f(\overline{X}_t)}{\sqrt{\overline{U}_t}}\right\|^2\right] + 2\sum_{t=1}^{T}\mathbb{E}\left[\left\|\frac{1}{N}\sum_{i=1}^{N}\frac{\nabla f_i(\overline{X}_t) - \nabla f_i(x_{t,i})}{\sqrt{\overline{U}_t}}\right\|^2\right]$$

and the last term on RHS of the above inequality can be bounded following similar procedures from (17) to (22), as what we did for $D_3$. Completing the procedures yields

$$\sum_{t=1}^{T}\mathbb{E}\left[\left\|\frac{1}{N}\sum_{i=1}^{N}\frac{\nabla f_i(\overline{X}_t) - \nabla f_i(x_{t,i})}{\sqrt{\overline{U}_t}}\right\|^2\right] \leq \sum_{t=1}^{T}\mathbb{E}\left[L\frac{1}{\epsilon}\frac{1}{N}\sum_{i=1}^{N}\|x_{t,i} - \overline{X}_t\|^2\right]$$

$$\leq \sum_{t=1}^{T}\mathbb{E}\left[L\frac{1}{\epsilon}\frac{1}{N}\alpha^2\left(\frac{1}{1-\lambda}\right)NdG_\infty^2\frac{1}{\epsilon}\right] \tag{33}$$

$$= TL\frac{1}{\epsilon^2}\alpha^2\left(\frac{1}{1-\lambda}\right)dG_\infty^2.$$

Finally, combining (30) to (33), we get

$$
\sum_{t=1}^{T} \mathbb{E}\left[\left\|\frac{1}{N}\sum_{i=1}^{N}\frac{g_{t,i}}{\sqrt{u_{t,i}}}\right\|^2\right] \leq 4\sum_{t=1}^{T}\mathbb{E}\left[\left\|\frac{\nabla f(\overline{X}_t)}{\sqrt{\overline{U}_t}}\right\|^2\right] + 4TL\frac{1}{\epsilon^2}\alpha^2\left(\frac{1}{1-\lambda}\right)dG_\infty^2
$$
$$
+ 2\frac{1}{\sqrt{N}}G_\infty^2\frac{1}{2\epsilon^2}\mathbb{E}\left[\sum_{o=0}^{T-1}\frac{\lambda}{1-\lambda}\|(-\hat{V}_{o-1}+\hat{V}_o)\|_{abs}\right] + T\frac{d}{N}\frac{\sigma^2}{\epsilon}
$$
$$
\leq 4\frac{1}{\sqrt{\epsilon}}\sum_{t=1}^{T}\mathbb{E}\left[\left\|\frac{\nabla f(\overline{X}_t)}{\overline{U}_t^{1/4}}\right\|^2\right] + 4TL\frac{1}{\epsilon^2}\alpha^2\left(\frac{1}{1-\lambda}\right)dG_\infty^2
$$
$$
+ 2\frac{1}{\sqrt{N}}G_\infty^2\frac{1}{2\epsilon^2}\mathbb{E}\left[\sum_{o=0}^{T-1}\frac{\lambda}{1-\lambda}\|(-\hat{V}_{o-1}+\hat{V}_o)\|_{abs}\right] + T\frac{d}{N}\frac{\sigma^2}{\epsilon}.
$$

where the last inequality is due to each element of $\overline{U}_t$ is lower bounded by $\epsilon$ by definition.

Combining all above, we obtain

$$
\frac{1}{T}\sum_{t=1}^{T}\mathbb{E}\left[\left\|\frac{\nabla f(\overline{X}_t)}{\overline{U}_t^{1/4}}\right\|^2\right]
$$
$$
\leq \frac{2}{T\alpha}(\mathbb{E}[f(Z_1)] - \mathbb{E}[f(Z_{T+1})])
$$
$$
+ \frac{L}{T}\alpha\left(\frac{\beta_1}{1-\beta_1}\right)^2\frac{G_\infty^2}{\sqrt{N}}\frac{1}{\epsilon^2}\frac{1}{1-\lambda}\mathbb{E}\left[\mathcal{V}_T\right]
$$
$$
+ \frac{8L}{T}\alpha\frac{1}{\sqrt{\epsilon}}\sum_{t=1}^{T}\mathbb{E}\left[\left\|\frac{\nabla f(\overline{X}_t)}{\overline{U}_t^{1/4}}\right\|^2\right] + 8L^2\alpha\frac{1}{\epsilon^2}\alpha^2\left(\frac{1}{1-\lambda}\right)dG_\infty^2 \qquad (34)
$$
$$
+ \frac{4L}{T}\alpha\frac{1}{\sqrt{N}}G_\infty^2\frac{1}{2\epsilon^2}\mathbb{E}\left[\sum_{o=0}^{T-1}\frac{\lambda}{1-\lambda}\|(-\hat{V}_{o-1}+\hat{V}_o)\|_{abs}\right] + 2L\alpha\frac{d}{N}\frac{\sigma^2}{\epsilon}
$$
$$
+ \frac{2}{T}\frac{\beta_1}{1-\beta_1}G_\infty^2\frac{1}{2\epsilon^{1.5}}\frac{1}{\sqrt{N}}\mathbb{E}\left[\frac{1}{1-\lambda}\mathcal{V}_T\right]
$$
$$
+ \frac{2}{T}\frac{G_\infty^2}{\sqrt{N}}\frac{1}{2\epsilon^{1.5}}\frac{\lambda}{1-\lambda}\mathbb{E}\left[\mathcal{V}_T\right]
$$
$$
+ \frac{3}{T}\left(\sum_{t=1}^{T}L\left(\frac{1}{1-\lambda}\right)^2\alpha^2 dG_\infty^2\frac{1}{\epsilon^{1.5}} + \sum_{t=1}^{T}L\left(\frac{\beta_1}{1-\beta_1}\right)^2\alpha^2 d\frac{G_\infty^2}{\epsilon^{1.5}}\right)
$$
$$
= \frac{2}{T\alpha}(\mathbb{E}[f(Z_1)] - \mathbb{E}[f(Z_{T+1})]) + 2L\alpha\frac{d}{N}\frac{\sigma^2}{\epsilon} + 8L\alpha\frac{1}{\sqrt{\epsilon}}\frac{1}{T}\sum_{t=1}^{T}\mathbb{E}\left[\left\|\frac{\nabla f(\overline{X}_t)}{\overline{U}_t^{1/4}}\right\|^2\right]
$$
$$
+ 3\alpha^2 d\left(\left(\frac{\beta_1}{1-\beta_1}\right)^2 + \left(\frac{1}{1-\lambda}\right)^2\right)L\frac{G_\infty^2}{\epsilon^{1.5}} + 8\alpha^3 L^2\left(\frac{1}{1-\lambda}\right)d\frac{G_\infty^2}{\epsilon^2}
$$
$$
+ \frac{1}{T\epsilon^{1.5}}\frac{G_\infty^2}{\sqrt{N}}\frac{1}{1-\lambda}\left(L\alpha\left(\frac{\beta_1}{1-\beta_1}\right)^2\frac{1}{\epsilon^{0.5}} + \lambda + \frac{\beta_1}{1-\beta_1} + 2L\alpha\frac{1}{\epsilon^{0.5}}\lambda\right)\mathbb{E}\left[\mathcal{V}_T\right].
$$

where $\mathcal{V}_T := \sum_{t=1}^{T}\|(-\hat{V}_{t-2}+\hat{V}_{t-1})\|_{abs}$. Set $\alpha = \frac{1}{\sqrt{dT}}$ and when $\alpha \leq \frac{\epsilon^{0.5}}{16L}$, we further have

$$
\frac{1}{T}\sum_{t=1}^{T}\mathbb{E}\left[\left\|\frac{\nabla f(\overline{X}_t)}{\overline{U}_t^{1/4}}\right\|^2\right]
$$
$$
\leq \frac{4}{T\alpha}(\mathbb{E}[f(Z_1)] - \mathbb{E}[f(Z_{T+1})]) + 4L\alpha\frac{d}{N}\frac{\sigma^2}{\epsilon}
$$

$$+ 6\alpha^2 d \left( \left( \frac{\beta_1}{1-\beta_1} \right)^2 + \left( \frac{1}{1-\lambda} \right)^2 \right) L \frac{G_\infty^2}{\epsilon^{1.5}} + 16\alpha^3 L^2 \left( \frac{1}{1-\lambda} \right) d \frac{G_\infty^2}{\epsilon^2}$$

$$+ \frac{2}{T\epsilon^{1.5}} \frac{G_\infty^2}{\sqrt{N}} \frac{1}{1-\lambda} \left( L\alpha \left( \frac{\beta_1}{1-\beta_1} \right)^2 \frac{1}{\epsilon^{0.5}} + \lambda + \frac{\beta_1}{1-\beta_1} + 2L\alpha \frac{1}{\epsilon^{0.5}} \lambda \right) \mathbb{E}\left[\mathcal{V}_T\right]$$

$$\leq \frac{4}{T\alpha} (\mathbb{E}[f(Z_1)] - \min_x f(x)) + 4L\alpha \frac{d}{N} \frac{\sigma^2}{\epsilon}$$

$$+ 6\alpha^2 d \left( \left( \frac{\beta_1}{1-\beta_1} \right)^2 + \left( \frac{1}{1-\lambda} \right)^2 \right) L \frac{G_\infty^2}{\epsilon^{1.5}} + 16\alpha^3 dL^2 \left( \frac{1}{1-\lambda} \right) \frac{G_\infty^2}{\epsilon^2}$$

$$+ \frac{2}{T\epsilon^{1.5}} \frac{G_\infty^2}{\sqrt{N}} \frac{1}{1-\lambda} \left( L\alpha \left( \frac{\beta_1}{1-\beta_1} \right)^2 \frac{1}{\epsilon^{0.5}} + \lambda + \frac{\beta_1}{1-\beta_1} + 2L\alpha \frac{1}{\epsilon^{0.5}} \lambda \right) \mathbb{E}\left[\mathcal{V}_T\right]$$

$$\leq C_1 \left( \frac{1}{T\alpha} (\mathbb{E}[f(Z_1)] - \min_x f(x)) + \alpha \frac{d\sigma^2}{N} \right) + C_2 \alpha^2 d + C_3 \alpha^3 d + \frac{1}{T\sqrt{N}} (C_4 + C_5 \alpha)\mathbb{E}\left[\mathcal{V}_T\right]$$

$$\tag{35}$$

where the first inequality is obtained by moving the term $8L\alpha \frac{1}{\sqrt{\epsilon}} \frac{1}{T} \sum_{t=1}^{T} \mathbb{E}\left[ \left\| \frac{\nabla f(\overline{X}_t)}{\overline{U}_t^{1/4}} \right\|^2 \right]$ on the RHS of (34) to the LHS to cancel it using the assumption $8L\alpha \frac{1}{\sqrt{\epsilon}} \leq \frac{1}{2}$ followed by multiplying both sides by 2. The constants introduced in the last step are defined as following

$$C_1 = \max(4, 4L/\epsilon),$$

$$C_2 = 6 \left( \left( \frac{\beta_1}{1-\beta_1} \right)^2 + \left( \frac{1}{1-\lambda} \right)^2 \right) L \frac{G_\infty^2}{\epsilon^{1.5}},$$

$$C_3 = 16L^2 \left( \frac{1}{1-\lambda} \right) \frac{G_\infty^2}{\epsilon^2},$$

$$C_4 = \frac{2}{\epsilon^{1.5}} \frac{1}{1-\lambda} \left( \lambda + \frac{\beta_1}{1-\beta_1} \right) G_\infty^2,$$

$$C_5 = \frac{2}{\epsilon^2} \frac{1}{1-\lambda} L \left( \frac{\beta_1}{1-\beta_1} \right)^2 G_\infty^2 + \frac{4}{\epsilon^2} \frac{\lambda}{1-\lambda} L G_\infty^2.$$

Substituting into $Z_1 = \overline{X}_1$ completes the proof. $\qquad\square$

## C    PROOF OF THEOREM 3

Under some assumptions stated in Corollary 2.1, we have that

$$\frac{1}{T} \sum_{t=1}^{T} \mathbb{E}\left[ \left\| \frac{\nabla f(\overline{X}_t)}{\overline{U}_t^{1/4}} \right\|^2 \right] \leq C_1 \frac{\sqrt{d}}{\sqrt{TN}} \left( (\mathbb{E}[f(Z_1)] - \min_x f(x)) + \sigma^2 \right) + C_2 \frac{N}{T} + C_3 \frac{N^{1.5}}{T^{1.5} d^{0.5}}$$

$$+ \left( C_4 \frac{1}{T\sqrt{N}} + C_5 \frac{1}{T^{1.5} d^{0.5}} \right) \mathbb{E}\left[ \sum_{t=1}^{T} \|(-\hat{V}_{t-2} + \hat{V}_{t-1})\|_{abs} \right] \tag{36}$$

where $\| \cdot \|_{abs}$ denotes the entry-wise $L_1$ norm of a matrix (i.e $\|A\|_{abs} = \sum_{i,j} |A_{ij}|$) and $C_1, C_2, C_3, C_4, C_5$ are defined in Theorem 2.

Since Algorithm 3 is a special case of 2, building on result of Theorem 2, we just need to characterize the growth speed of $\mathbb{E}\left[ \sum_{t=1}^{T} \|(-\hat{V}_{t-2} + \hat{V}_{t-1})\|_{abs} \right]$ to prove convergence of Algorithm 3. By the

update rule of Algorithm 3, we know $\hat{V}_t$ is non decreasing and thus

$$
\mathbb{E}\left[\sum_{t=1}^{T}\|(-\hat{V}_{t-2}+\hat{V}_{t-1})\|_{abs}\right] = \mathbb{E}\left[\sum_{t=1}^{T}\sum_{i=1}^{N}\sum_{j=1}^{d}|-[\hat{v}_{t-2,i}]_j+[\hat{v}_{t-1,i}]_j|\right]
$$

$$
= \mathbb{E}\left[\sum_{t=1}^{T}\sum_{i=1}^{N}\sum_{j=1}^{d}(-[\hat{v}_{t-2,i}]_j+[\hat{v}_{t-1,i}]_j)\right]
$$

$$
= \mathbb{E}\left[\sum_{i=1}^{N}\sum_{j=1}^{d}(-[\hat{v}_{-1,i}]_j+[\hat{v}_{T-1,i}]_j)\right]
$$

$$
= \mathbb{E}\left[\sum_{i=1}^{N}\sum_{j=1}^{d}(-[\hat{v}_{0,i}]_j+[\hat{v}_{T-1,i}]_j)\right],
$$

where the last equality is because we defined $\hat{V}_{-1} \triangleq \hat{V}_0$ previously.

Further, because $\|g_{t,i}\|_\infty \le G_\infty$ for all $t,i$ and $v_{t,i}$ is a exponential moving average of $g_{k,i}^2$, $k = 1, 2, \cdots, t$, we know $|[v_{t,i}]_j| \le G_\infty^2$, for all $t, i, j$. In addition, by update rule of $\hat{V}_t$, we also know each element of $\hat{V}_t$ also cannot be greater than $G_\infty^2$, i.e. $|[\hat{v}_{t,i}]_j| \le G_\infty^2$, for all $t, i, j$. Given the fact that $[\hat{v}_{0,i}]_j \ge 0$, we have

$$
\mathbb{E}\left[\sum_{t=1}^{T}\|(-\hat{V}_{t-2}+\hat{V}_{t-1})\|_{abs}\right] = \mathbb{E}\left[\sum_{i=1}^{N}\sum_{j=1}^{d}(-[\hat{v}_{0,i}]_j+[\hat{v}_{T-1,i}]_j)\right] \le \mathbb{E}\left[\sum_{i=1}^{N}\sum_{j=1}^{d}G_\infty^2\right] = NdG_\infty^2.
$$

Substituting the above into (39), we have

$$
\frac{1}{T}\sum_{t=1}^{T}\mathbb{E}\left[\left\|\frac{\nabla f(\overline{X}_t)}{\overline{U}_t^{1/4}}\right\|^2\right] \le C_1\frac{\sqrt{d}}{\sqrt{TN}}\left((\mathbb{E}[f(Z_1)]-\min_x f(x))+\sigma^2\right)+C_2\frac{N}{T}+C_3\frac{N^{1.5}}{T^{1.5}d^{0.5}}
$$

$$
+\left(C_4\frac{1}{T\sqrt{N}}+C_5\frac{1}{T^{1.5}d^{0.5}}\right)NdG_\infty^2
$$

$$
= C_1'\frac{\sqrt{d}}{\sqrt{TN}}\left((\mathbb{E}[f(Z_1)]-\min_x f(x))+\sigma^2\right)+C_2'\frac{N}{T}+C_3'\frac{N^{1.5}}{T^{1.5}d^{0.5}}
$$

$$
+C_4'\frac{\sqrt{N}d}{T}+C_5'\frac{Nd^{0.5}}{T^{1.5}},
$$

$$
\tag{37}
$$

where we have

$$
C_1' = C_1 \quad C_2' = C_2 \quad C_3' = C_3 \quad C_4' = C_4 G_\infty^2 \quad C_5' = C_5 G_\infty^2. \tag{38}
$$

and we conclude the proof. $\qquad\square$

## D    PROOF OF THEOREM 4

The proof follows the same flow as that of Theorem 3. Under assumptions stated in Corollary 2.1, set $\alpha = \sqrt{N}/\sqrt{Td}$, we have that

$$
\frac{1}{T}\sum_{t=1}^{T}\mathbb{E}\left[\left\|\frac{\nabla f(\overline{X}_t)}{\overline{U}_t^{1/4}}\right\|^2\right] \le C_1\frac{\sqrt{d}}{\sqrt{TN}}\left((\mathbb{E}[f(Z_1)]-\min_x f(x))+\sigma^2\right)+C_2\frac{N}{T}+C_3\frac{N^{1.5}}{T^{1.5}d^{0.5}}
$$

$$
+\left(C_4\frac{1}{T\sqrt{N}}+C_5\frac{1}{T^{1.5}d^{0.5}}\right)\mathbb{E}\left[\sum_{t=1}^{T}\|(-\hat{V}_{t-2}+\hat{V}_{t-1})\|_{abs}\right],
$$

$$
\tag{39}
$$

where $\|\cdot\|_{abs}$ denotes the entry-wise $L_1$ norm of a matrix (i.e $\|A\|_{abs} = \sum_{i,j} |A_{ij}|$) and $C_1, C_2, C_3, C_4, C_5$ are defined in Theorem 2.

Again, Since decentralized AdaGrad is a special case of 2, we can apply Corollary 2.1 and what we need is to upper bound $\mathbb{E}\left[\sum_{t=1}^T \|(-\hat{V}_{t-2} + \hat{V}_{t-1})\|_{abs}\right]$ derive convergence rate. By the update rule of decentralized AdaGrad, we have $\hat{v}_{t,i} = \frac{1}{t}(\sum_{k=1}^t g_{k,i}^2)$ for $t \geq 1$ and $\hat{v}_{0,i} = \epsilon \mathbf{1}$. Then we have for $t \geq 3$,

$$\mathbb{E}\left[\sum_{t=1}^T \|(-\hat{V}_{t-2} + \hat{V}_{t-1})\|_{abs}\right]$$

$$=\mathbb{E}\left[\sum_{t=1}^T \sum_{i=1}^N \sum_{j=1}^d |-[\hat{v}_{t-2,i}]_j + [\hat{v}_{t-1,i}]_j|\right]$$

$$\leq\mathbb{E}\left[\sum_{t=3}^T \sum_{i=1}^N \sum_{j=1}^d |-\frac{1}{t-2}([\sum_{k=1}^{t-2} g_{k,i}^2]_j) + \frac{1}{t-1}([\sum_{k=1}^{t-1} g_{k,i}^2]_j)|\right] + Nd(G_\infty^2 - \epsilon)$$

$$\leq\mathbb{E}\left[\sum_{t=3}^T \sum_{i=1}^N \sum_{j=1}^d |(\frac{1}{t-1} - \frac{1}{t-2})([\sum_{k=1}^{t-2} g_{k,i}^2]_j) + \frac{1}{t-1}[g_{t-1,i}^2]_j)|\right] + NdG_\infty^2$$

$$=\mathbb{E}\left[\sum_{t=3}^T \sum_{i=1}^N \sum_{j=1}^d |(-\frac{1}{(t-1)(t-2)})([\sum_{k=1}^{t-2} g_{k,i}^2]_j) + \frac{1}{t-1}[g_{t-1,i}^2]_j|\right] + NdG_\infty^2$$

$$\leq\mathbb{E}\left[\sum_{t=3}^T \sum_{i=1}^N \sum_{j=1}^d \max\left(\frac{1}{(t-1)(t-2)}([\sum_{k=1}^{t-2} g_{k,i}^2]_j), \frac{1}{t-1}[g_{t-1,i}^2]_j\right)\right] + NdG_\infty^2$$

$$\leq\mathbb{E}\left[Nd\sum_{t=3}^T \frac{G_\infty^2}{t-1}\right] + NdG_\infty^2$$

$$\leq NdG_\infty^2 \log(T) + NdG_\infty^2$$

$$=NdG_\infty^2(\log(T) + 1)$$

where the first equality is because we defined $\hat{V}_{-1} \triangleq \hat{V}_0$ previously and $\|g_{k,i}\|_\infty \leq G_\infty$ by assumption.

Substituting the above into (39), we have

$$\frac{1}{T}\sum_{t=1}^T \mathbb{E}\left[\left\|\frac{\nabla f(\overline{X}_t)}{\overline{U}_t^{1/4}}\right\|^2\right] \leq C_1 \frac{\sqrt{d}}{\sqrt{TN}}\left((\mathbb{E}[f(Z_1)] - \min_x f(x)) + \sigma^2\right) + C_2 \frac{N}{T} + C_3 \frac{N^{1.5}}{T^{1.5}d^{0.5}}$$

$$+ \left(C_4 \frac{1}{T\sqrt{N}} + C_5 \frac{1}{T^{1.5}d^{0.5}}\right) NdG_\infty^2(\log(T) + 1)$$

$$= C_1' \frac{\sqrt{d}}{\sqrt{TN}}\left((\mathbb{E}[f(Z_1)] - \min_x f(x)) + \sigma^2\right) + C_2' \frac{N}{T} + C_3' \frac{N^{1.5}}{T^{1.5}d^{0.5}}$$

$$+ C_4' \frac{d\sqrt{N}(\log(T) + 1)}{T} + C_5' \frac{(\log(T) + 1)N\sqrt{d}}{T^{1.5}},$$

where we have

$$C_1' = C_1 \quad C_2' = C_2 \quad C_3' = C_3 \quad C_4' = C_4 G_\infty^2 \quad C_5' = C_5 G_\infty^2. \tag{40}$$

and we conclude the proof. $\qquad\square$

# E   ADDITIONAL EXPERIMENTS AND DETAILS

In this section, we compare the training loss and testing accuracy of different algorithms, namely Decentralized Stochastic Gradient Descent (DGD), Decentralized Adam (DADAM) and our proposed Decentralized AMSGrad, with different stepsizes on heterogeneous data distribution. We use 5 nodes and the heterogeneous data distribution is created by assigning each node with data of only two labels. Note that there are no overlapping labels between different nodes. For all algorithms, we compare stepsizes in the grid $[10^{-1}, 10^{-2}, 10^{-3}, 10^{-4}, 10^{-5}, 10^{-6}]$.

Figure 2 shows the training loss and test accuracy for DGD algorithm. We observe that the stepsize $10^{-3}$ works best for DGD in terms of test accuracy and $10^{-1}$ works best in terms of training loss. This difference is caused by the inconsistency among the value of parameters on different nodes when the stepsize is large. The training loss is calculated as the average of the loss value of different local models evaluated on their local training batch. Thus, while the training loss is small at a particular node, the test accuracy will be low when evaluating data with labels not seen by the node (recall that each node contains data with different labels since we are in the heterogeneous setting).

Figure 3 shows the performance of decentralized AMSGrad with different stepsizes. We see that its best performance is better than the one of DGD and the performance is more stable (the test performance is less sensitive to stepsize tuning).

Figure 4 displays the performance of Decentralized Adam algorithm. As expected, the performance of DADAM is not as good as DGD or decentralized AMSGrad. Its divergence characteristic, highlighted Section 2.3, coupled with the heterogeneity in the data amplify its non-convergence issue in our experiments. From the experiments above, we can see the advantages of decentralized AMSGrad in terms of both performance and ease of parameter tuning, and the importance of ensuring the theoretical convergence of any newly proposed methods in the presented setting.

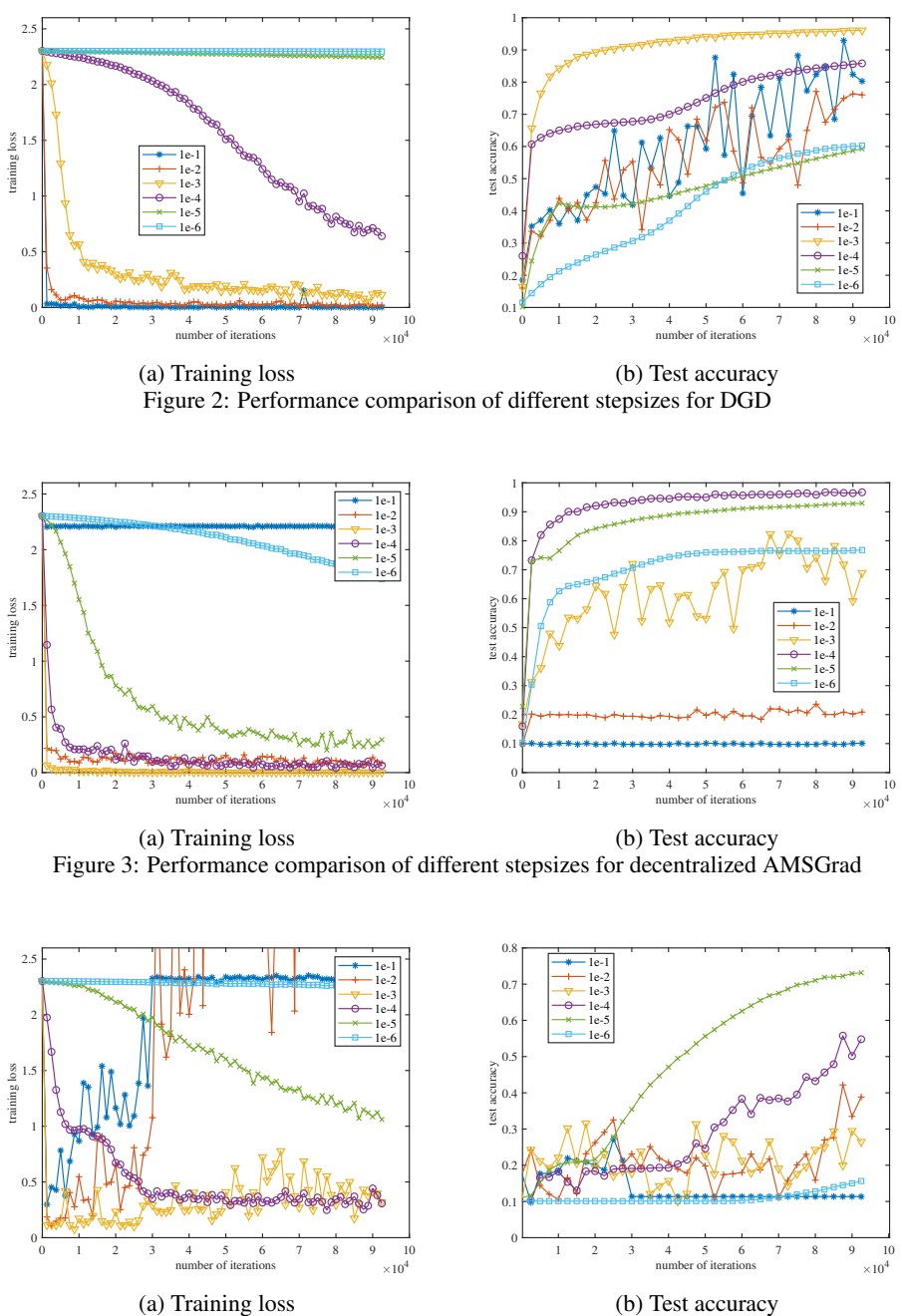

(a) Training loss           (b) Test accuracy

Figure 2: Performance comparison of different stepsizes for DGD

(a) Training loss           (b) Test accuracy

Figure 3: Performance comparison of different stepsizes for decentralized AMSGrad

(a) Training loss           (b) Test accuracy

Figure 4: Performance comparison of different stepsizes for DADAM

