# OpenReview forum: "Convergent Adaptive Gradient Methods in Decentralized Optimization"
_ICLR.cc/2021/Conference — Reject_

### Official Review · AnonReviewer2 · 2020-10-20
**This paper  is very confusing and not well-written. It  cannot be accepted.**

**Rating:** 3
**Confidence:** 3

**Review:**

This paper consider the decentralized adaptive algorithms. At the first glance,  I am really happy to that the adaptive methods are used for the decentralized optimization. However,  after I read the  main document, I do not think  paper actually    analyzes the decentralized adaptive algorithms.

In line 9 of Algorithm 1, the   denominator is $\sqrt{\hat{v}_{t,i}}$.

 However, in Algorithms 2 and 3, it is changed as $\sqrt{u_{t,i}}$. In  the proofs, the authors proved the convergence based on $u_{t,i}\geq \epsilon$. This is actually the DSGD. The proofs can be quite simple.  And the restriction $\alpha=O(\sqrt{\epsilon})$ can be easily removed.
This paper does not present any insights for the decentralized  adaptive methods. It only depends on $u_{t,i}\geq \epsilon$. The numerical results show that the proposed method is similar as DSGD. As mentioned before, it is actually DSGD but with slightly modification.

---

> ### Author Response · Authors · 2020-11-19
> **We respectfully disagree with the remark that our analysis only covers "a slightly modified DSGD"**
>
> We thank the reviewer for the comments on our contribution.
>
> However, we respectfully disagree with the remark regarding the fact that our analysis covers "a slightly modified DSGD".
>
> Essentially, Algorithm 2 is not DSGD but rather a decentralized framework of some adaptive gradient methods (depending on the choice of the function $r_t$ in line 6). The change of notation from $\hat v_{t,i}$ to $u_{t,i}$ is simply because in our decentralized version of adaptive methods, unlike DADAM of [Nazari et. al., 2019], the algorithm (Alg. 2) uses the average consensus of $\hat v_{t,i}$ to help convergence (recall that DADAM fails to converge in some settings). Hence, $\hat v_{t,i}$ is not simply "changed to" $u_{t,i}$ but converted into an auxiliary quantity noted $u_{t,i}$ that is the result of several operations, see line 8 and 11, on the original $\hat{v}_{t,i}$.
>
> Our analysis and algorithms are truly with respect to decentralized variants of adaptive methods. Consequently, the proofs for our newly proposed decentralized framework differ from those for DSGD and we invite the reviewer to check them in the appendix. We will change the notation of $u_{t,i}$ to avoid any confusion.

---

### Official Review · AnonReviewer1 · 2020-10-22
**Everything seems sound from first glance, but I did not have a chance to carefully check the proofs**

**Rating:** 7
**Confidence:** 1

**Review:**

This paper discusses the problem of adaptive acceleration in the decentralized setting. The premise is that decentralized versions, while successful in the simple SGD setting, do not extend well in the acceleration settings, e.g. Adam and Adagrad. They first present a counterexample where a simple decentralized scheme, applying Adam, converges to a nonstationary point.  (Suggestion: I would maybe add a cleaner, more flushed out version of this proof in the appendix, maybe with illustrations.)

Overall, everything the paper presented seemed reasonable. The motivation and counterexample in DADAM case are solid, and the following theorems seem to suggest gradient error norm $\to 0$ at rate $O(1/\sqrt{T})$ which is reasonable in nonconvex optimization.  The intuition in the adjusted merging scheme is also reasonable, and makes sense that it would work better than vanilla merging schemes. However, I did not have a chance to carefully check the proofs, which are clearly the main contribution of the paper.

One thing I would suggest is a more thorough set of numerical experiments. The two examples shown, in fact all the methods converge, and while the proposed method converges faster, it isn't really verifying the paper's main point, which is that the standard distributed methods diverge and the proposed method converges. Showing this on a standard machine learning task would improve motivation.

---

> ### Author Response · Authors · 2020-11-19
> **Thank you for the suggestions, we will revise our paper soon**
>
> While we acknowledge that the numerical experiments can be improved by adding runs on larger datasets (we will try to do so during the rebuttal period), we want to explain to the reviewer that the current experiments support our Theory. The current experiments we are displaying in our paper are informative on how our newly proposed decentralized framework behaves with respect to baseline methods.
> In Figure 1 (b), we show a very bad convergence behavior of DADAM on heterogeneous data, in echo of the theoretical divergence that we claim.
> Nevertheless, our decentralized framework, here adapted to the specific AMSGrad method, and D-PSGD (the algorithm in Lian et. al. is actually called D-PSGD, we will fix that in the revision) are exhibiting great convergence. our framework is similar and sometimes better than D-PSGD.
> While D-PSGD is a non-adaptive decentralized method, Figure 1 is convincing on the need for a convergent decentralized *adaptive* method, thus fixing the divergence issue of DADAM (shown both theoretically and empirically through Figure 1).

---

### Official Review · AnonReviewer4 · 2020-10-28
**A well-written paper that addresses an improtant problem in distributed learning literature**

**Rating:** 8
**Confidence:** 4

**Review:**

The paper introduces a decentralized framework for adaptive momentum-based gradient descent optimizers, such as ADAM. The proposed method is novel and is among the first works to consider a decentralized communication graph without a master node. The author discovers the divergent properties of the recent work of DADAM (Nazari, 2019) and proposes a way to fix it by adding a similar consensus step for the adaptive learning rates of agents. The mathematical derivation seems to be correct to the best of my knowledge. Finally, the author tests their method on a simple CNN and show their superiority compared to DADAM to achieve a close to the centralized performance.

However, I have some minor comments to improve the manuscript.
1. The experimental evaluation of the work is quite limited. I understand the space limit but it would have been nice to see more experiments instead of showcasing Algorithm 2 with an extra example in Algorithm 3. It is important to see the convergence behavior of the method (on the training data) with respect to the DGD on various datasets/networks in practice, rather than observing how the testing accuracy behaves. Note that your method does not guarantee any specific generalization behavior and therefore I believe it is more suited to report the experiments only in terms of training performance when you are out of space.

2. What are the drawbacks of this method? I can see more memory requirements for the agents due to the new variable \tilde{u} for instance. Do you have any quantified evaluation in this respect? I suspect it can be significant especially if the trained model is large and the agents have limited memory/computational resources

3. In Section 3.2, the author says "Algorithm 2 can become different adaptive gradient methods by specifying r_t as different functions. E.g., when we choose ..., Algorithm 2 becomes a decentralized version of AdaGrad." This sentence is not accurate as algorithms like AdaGrad and Adadelta do not use momentum on the past gradients. They only use the squared values of the past gradients. I believe your method, as I mentioned above, is a general framework for momentum-based techniques including ADAM, AdaMax, NADAM, etc, which brings me to the next question.

4. Is it possible to generalized your method for an adaptive gradient descent algorithm that does not use the momentum of the gradients? For example, take AdaGrad with a fixed learning rate of \eta instead of m_t. How does your convergence behavior change?

---

> ### Author Response · Authors · 2020-11-19
> **Thank you for the valuable suggestions, we will revise accordingly during the rebuttal period**
>
> We thank you for the valuable comments on our submission. We are revising our paper and will update as soon as it is done. Following is our answer to your questions.
>
>
> 1. Numerical experiments plots
>
>
> We agree with the reviewer that experiments on various datasets/networks will be a good improvement to our paper, we will try to add more experiments in a future version. What we want to highlight in the experiments is that in Figure 1 (b), DADAM performs bad in terms of both training and testing performance, while decentralized AMSGrad performs as good as D-PSGD (even better). This shows the divergence issue for DADAM can have significant impact on model performance in practice instead of being a negligible defect, supporting the importance of designing theoretically convergent algorithms.
>
>
> 2. What are the drawbacks of this method?
>
> As noticed by the reviewer, one drawback of our framework is the additional memory requirement for $\tilde u_{t,i}$. In our framework, across iterations, each node have to save a few quantities including $m_{t,i}$, $x_{t,i}$, memories for $\hat v_{t,i}$, and $\\tilde u_{t,i}$. Take decentralized AMSGrad (Algorithm 3) for an example, we  need $d$ floats for  $m_{t,i}$, $2d$ floats for $\hat v_{t,i}$ (because both $v_{t,i}$ in line 6 and $\hat v_{t,i}$ in line 7 of Algorithm 3 requires $d$ floats), $d$ floats for $x_{t,i}$, and additional $d$ floats for $\tilde u_{t,i}$. Overall, we need to store $5d$ floats while centralized AMSGrad requires storing $4d$ floats (without $\tilde u_{t,i}$).  Thus, decentralized AMSGrad requires $25\%$ more storage compared with centralized AMSGrad which is acceptable in certain situations. However, compared with only $d$ floats storage requirement in D-PSGD, decentralized AMSGrad indeed introduced significantly more memory.  We also want to remark that the memory requirement can be smaller if the memory for adaptive learning rate $\hat v_{t,i}$ is smaller for other possible adaptive gradient methods. Suppose $k$ instead of $d$ floats are used for $\hat v_{t,i}$ (e.g. per-layer adaptive learning rate, $k$ being the number of layers). One may also use $k$ floats for $\tilde u_{t,i}$, leading to a memory requirement of $2d+2k$ floats.
>
> Another drawback of our method is the doubled amount of communication compared with DADAM and D-PSGD since we need consensus operation for $\tilde u_{t,i}$ in addition to $x_{t,i}$.
>
>
> 3. Is it possible to generalize the method to adaptive gradient methods without momentum such as AdaGrad?
>
> It is possible to analyze the convergence of adaptive gradient without momentum using our framework. Actually a decentralized version of AdaGrad falls into our framework by setting $\hat v_{t,i}  = \frac{1}{t}\sum_{k=1}^t g_{k,i}^2$ and $\beta_1 = 0$. Then  Theorem 2 applies and then we can focus on bounding the term $\\mathbb E [\\sum _{t=1}^T \\| (-\\hat V _{t-2} + \\hat V _{t-1}) \\| _{abs}]$ in (2) for this algorithm. Finally, some algebraic manipulations can show $\\mathbb E [\\sum _{t=1}^T \\| (-\\hat V _{t-2} + \\hat V _{t-1}) \\| _{abs}] = O(Nd\log(T))$ for this algorithm, which is worse than decentralized AMSGrad by a factor of $O(\log(T))$ on this term. Substituting this bound into (2) we can see that this decentralized version of AdaGrad also converges with a rate of $O(\frac{\sqrt{d}}{\sqrt{T}})$

---

> > ### Author Response · Authors · 2020-11-24
> > **Results for AdaGrad are added.**
> >
> > We want to thank you again for the valuable suggestions. In response to the reviewer's questions, we have generalized our framework to design a decentralized version of AdaGrad in Section 4. Theory shows that this algorithm can also achieve the $O(1/\sqrt{T})$ convergence rate.

---

### Official Review · AnonReviewer3 · 2020-10-28
**This paper still need many improvements.**

**Rating:** 4
**Confidence:** 5

**Review:**

In this paper, the authors attempt to use adaptive gradient methods in decentralized training paradigm. They develop a general framework to convert an adaptive gradient method from a centralized one to its decentralized variant. Specifically, they propose a decentralized AMSGrad algorithm. They also point out a potential divergent problem of an existing method and investigate the conditions to ensure convergence. Finally, they conduct some experiments to verify the performance of their algorithm.

Pros:
1. This paper is the first one to use adaptive gradient method to decentralized training paradigm and the authors also provide a completed convergence analysis for their algorithm.
2. This paper reveals a divergent problem of DADAM in offline situation by taking some intuitive examples. Based on this issue, they find a solution that makes consensus on the adaptive learning rates.
3. This paper proposes a general framework to apply adaptive gradient methods to decentralized optimization. The authors also investigate the conditions to make sure these decentralized variants will converge.

Cons:
1. In section 3.2, the paper claims AdaGrad and AMSGrad satisfy the condition to guarantee the convergence of Algorithm 2 while Adam does not. It seems not to be an obvious conclusion from the reference Chen et al. (2019). Is there more explanation or proof about why AdaGrad and AMSGrad satisfy the condition? And does it mean Adam still diverges even after using the algorithmic approach proposed in this paper?
2. In Theorem 2, the convergence analysis result of Algorithm 2 is given. However, the convergence of common adaptive gradient methods such as AdaGrad and Adam is still not clear. Therefore, the “convergent adaptive gradient method” in the title is very misleading. Do most of the adaptive gradient methods have the same theoretical guarantees as AMSGrad?
3. \mathbb{E}[\sum_{t=1}^T \lVert (-\hat{V}_{t-2} + \hat{V}_{t-1}) \rVert_{abs}] = o(T) is the key condition to ensure the convergence. But when the above equation is O(\sqrt{T}), the convergence rate is worse than the centralized counterpart. Is that case possible?
4. In section 3.4, the experiment is divided into homogeneous and heterogeneous data, which is very confusing. What is the reason for doing this and what will happen if we just deal with the dataset normally? The heterogeneous data is treated very intentionally. Is there any discussion about when the treatment of heterogeneous data is important?
5. In the homogeneous data experiment, the performance of DADAM and decentralized AMSGrad are similar. What is the reason that the learning rates on different node tend to be similar? Is that a common case? Maybe the experiment on more dataset is needed to address this concern. Besides, how will such similarity among data impact the theoretical convergence?

Minors:
The algorithm proposed in Lian et al. (2017) is called Decentralized Parallel Stochastic Gradient Descent (D-PSGD).

---

> ### Author Response · Authors · 2020-11-19
> **We are revising our paper based on your comments and will update soon**
>
> Thank you for the questions and suggestions. We are revising our paper based on your comments and will update when the revision is done. We provide point-to-point answers to your questions below.
>
> 1. More explanation or proof about why AdaGrad and AMSGrad satisfy the condition? Does Adam still diverge after the algorithmic approach?
>
> AdaGrad and AMSGrad can satisfy the condition essentially due to the stability in adaptive learning rate in their update rules. The sequence of adaptive learning rate in vanilla AMSGrad is non-decreasing, thus the overall oscillation of adaptive learning across T iterations (i.e. the term $ \\mathbb{E}[\\sum_{t=1}^T  \\|-\\hat V _{t-2} + \\hat V _{t-1}\\| _{abs} ] $ in (2)) is actually bounded by a constant. For AdaGrad, its adaptive learning rate can be viewed as the average of all past squared gradients (coordinate-wise). Due to the average, as $t$ grows, the difference between adjacent adaptive learning rate is getting smaller. This ensures that the overall oscillation of adaptive learning rate in AdaGrad is bounded by $O(\\log(T))$.
>
> For Adam, it will still diverge even after the algorithmic approach. This is because the original Adam is divergent, our algorithmic approach can only convert certain convergent adaptive algorithms to their decentralized counterparts. However, Adam is not a convergent algorithm as proven in [Reddi et al., 2018].  As indicated by Theorem 2, the last term (oscillation of adaptive learning rate) in (2) should be o(T) to ensure convergence.  Yet, the term is $O(T)$ for Adam in worst case, which is unbounded.
>
> 2. Convergence of common adaptive gradient methods such as AdaGrad and Adam is not clear in Theorem 2, thus the title is misleading. Do most of the adaptive gradient methods have the same theoretical guarantees as AMSGrad?
>
>
> As noticed by the reviewer, Theorem 2 highlights a condition to ensure convergence of adaptive gradient methods converted by our framework, i.e. $ \\mathbb{E}[\\sum_{t=1}^T  \\|-\\hat V _{t-2} + \\hat V _{t-1}\\| _{abs} ]  = o(T)$. Such a condition also appears in [Chen et al., 2019] and  is satisfied by AMSGrad and AdaGrad, but not Adam. Thus, AdaGrad will converge but Adam will not. We will add more detailed discussion on this condition for AMSGrad, AdaGrad, and Adam below Theorem 2.
>
>
> 3. When $ \mathbb{E}[\\sum _{t=1}^T  \\|-\\hat V _{t-2} + \\hat V _{t-1}\\| _{abs} ] $ is larger than $O(\\sqrt{T})$, the convergence rate is worse than the centralized counterpart.  $ \mathbb{E}[\\sum _{t=1}^T  \\|-\\hat V _{t-2} + \\hat V _{t-1}\\| _{abs} ] $
>
> In short, such a situation is possible, yet, to the best of our knowledge, the term  $ \mathbb{E}[\\sum _{t=1}^T  \\|-\\hat V _{t-2} + \\hat V _{t-1}\\| _{abs} ] $ for many convergent adaptive algorithms is smaller than $O(\sqrt{T})$. Specifically, based on [Reddi et al., 2018, Chen et al., 2019], it can be proven that AdaGrad and AMSGrad satisfy $ \mathbb{E}[\\sum _{t=1}^T  \\|-\\hat V _{t-2} + \\hat V _{t-1}\\| _{abs} ]$ being $O(Nd)$ and $O(Nd\log(T))$, respectively.  In addition, the convergence rate of centralized AMSgrad and AdaGrad actually also depends on a centralized counterpart of this term. Thus, our bound will not be worse than the centralized counterparts for these algorithms. We will add a comment on this below Theorem 2.
>
> However, we agree with the reviewer that it is possible for convergent algorithms to have $ \mathbb{E}[\\sum _{t=1}^T  \\|-\\hat V _{t-2} + \\hat V _{t-1}\\| _{abs} ] $ being large or unbounded, even $O(T)$. For example, when the adaptive learning rate is a positive random number generated independently at each iteration, we can have $ \mathbb{E}[\\sum _{t=1}^T  \\|-\\hat V _{t-2} + \\hat V _{t-1}\\| _{abs} ]  = O(T)$. Yet, because the adaptive learning rate does not depend on gradients, the convergence bias caused by correlation between adaptive learning rate and stochastic gradient is 0 and the algorithm can converge.
>
> 4. Reason for dividing homogeneous data and heterogeneous data.
>
> We consider these two settings separately because both of them can happen in real world applications and the performance of different algorithms could be significantly different on these settings. Following are real world examples of homogeneous and heterogeneous data distributions.
>
>    1). Homogeneous data: The dataset is shuffled and uniformly randomly distributed on different nodes. Or all the nodes have access to the same dataset. These settings are common for computer clusters.
>
>    2). Heterogeneous data: Each node will collect their own dataset, the generating data distributions of different nodes are different. A data shuffling is prohibited because the nodes are reluctant to share data. Such settings are common in a cooperative training situations, one notable setting is federated learning.

---

> > ### Author Response · Authors · 2020-11-19
> > **continued response**
> >
> > 5. Why DADAM and decentralized AMSGrad are similar in the homogeneous data experiment? Is that a common case? How will similarity among data impact the theoretical convergence?
> >
> > These two algorithms are similar because the difference between DADAM and decentralized AMSGrad is the adaptive learning rate. DADAM uses individual adaptive learning rate on different nodes while decentralized AMSGrad uses approximate average of adaptive learning rate across workers in DADAM as the true adaptive learning rate. As mentioned in the paper, the divergence of DADAM is caused by the difference of adaptive learning rate on differnet nodes. If the difference is small or even 0, DADAM should not perform too bad. On homogeneous data distribution, the individual adaptive learning rates will be similar to the average of them since the gradient distribution on different nodes are similar.
> > The impact of distribution similarity on different nodes on the convergence of DADAM is a very interesting research question but we found such a quantitative analysis on  it to be non-trivial.
> > We will leave this for future work.

---

> > > ### Author Response · Authors · 2020-11-24
> > > **More explanations and intuitions are added, results extended to AdaGrad**
> > >
> > > Based on the reviewer's comments, we revised the discussions in our paper. Specifically,
> > > 1. We added discussions on the convergence rate for AMSGrad and AdaGrad below Corollary 2.1 (original Theorem 2). More intuitions are provided.
> > > 2. We added discussion on example settings for homogeneous and heterogeneous data distribution (page 9).
> > > 3. We also extended our framework to AdaGrad, provided a decentralized version of AdaGrad with convergence analysis.

---

### Official Review · AnonReviewer5 · 2020-11-06
**Weak theoretical results with weak experiments.**

**Rating:** 3
**Confidence:** 5

**Review:**


This paper studied the decentralized adaptive gradient methods and provided convergence guarantees. Experiment on MNIST is conducted to show the effectiveness of the proposed approach.

1. The theoretical result is weak. The linear speedup result is not proved as in (Lian et al. 2017), the benefits of adaptive gradient methods are also not illustrated in the bound in Theorem 2 and Theorem 3.

2. The learning rate scheme is not practical and does not hold in practice. As illustrated in Theorem 2 and 3, the learning rate $\alpha$ is set to be less than $\epsilon^{0.5}/16L$.

3. The LHS of Theorem 2 and Theorem 3 are not the standard gradient squared norm but the scaled version. It is unclear what is the bound if the LHS is the standard gradient squared norm as in (Lian et al. 2017). It is important to use the same measure as in the previous literature for fair comparison.

3. The experiment is weak. Doing distributed training only on a tiny dataset on MNIST is not sufficient. I would like to see results on larger datasets such as CIFAR and ImageNet.

---

> ### Author Response · Authors · 2020-11-19
> **We will add linear speedup results and will update as soon as all revisions are done.**
>
> We thank the reviewer for the comments/remarks on our paper.
> We wish to provide as many clarifications as needed to remove any doubts.
>
> 1. Linear speedup is not proved, the benefits of adaptive gradient methods are not shown in theory.
>
> Regarding the linear speedup as in Lian et.al, we will add in the rebuttal version, a similar bound for our decentralized scheme. Thank you for the suggestion. While the advantage of the adaptive gradient methods does not show in our bound, we must stress that this latter bound showcases similar complexity in terms of number of iterations as centralized adaptive methods.
> Thus, this unclear advantage is inherited by current bounds in the centralized paradigm. Although we believe that theoretically showing that adaptive methods present a clear edge over non adaptive methods is an interesting and important research problem, we emphasize on the contribution of our paper being on the derivation of a general framework to convert certain adaptive gradient methods into their decentralized counterparts. The theoretical results we present aim at statistically prove convergence of this framework instead of showing benefits.
>
> 2. The learning rate less than $\epsilon^{0.5}/16L$ is not practical.
>
> In our paper, the theoretical contributions focus on showing the convergence guarantee in terms of $T$ since even such basic guarantee is not guaranteed by DADAM. The assumption that step size is smaller than $\epsilon^{0.5}/16L$ can be relaxed by adding another $O(\frac{1}{\sqrt{T}})$ error term on RHS of (2). We will add a comment on this stepsize requirement below Theorem 2.
>
> 3. Relate LHS of Theorem 2 and 3 to standard gradient squared norm.
>
> The LHS of Theorem 2 and 3 can be translated into standard gradient bound by using worst-case $L_{\infty}$ upper on $\overline U_{t}$, which depends on $\hat v_{t,i}$.  This can not be specified in Theorem 2 since it is a framework and $\hat v_{t,i}$ is rather general and not explicit, making our result more broad.  For Theorem 3, we can have all coordinates of $\overline U_{t}$ being less than or equal to $G_{\infty}^2$ (due to update rules for $\hat v_{t,i}$) and thus $\|\frac{\nabla f(\overline X_t)}{\overline U_t^{1/4}}\|^2 \geq \frac{1}{G_{\infty}}\|{\nabla f(\overline X_t)}\|^2$. Then this measure can be translated into standard gradient norm bound. We will add discussion on this below Theorem 3. We want to friendly remind the reviewer that the main focus of this work is not to prove matching bounds with (Lian et al. 2017) but to provide a framework for designing convergent decentralized adaptive gradient methods while DADAM diverges.
>
>
> 4. The experiment is weak.
>
> We agree that experiments on larger datasets will be a good improvement to our paper, we will try to add more experiments in the rebuttal. Yet, our current experiment shows a clear advantage of decentralized AMSGrad over DADAM on heterogeneous dataset. In Figure 1 (b), DADAM performs really bad in both training and testing compared with DGD, caused by its possible divergence. In contrast, decentralized AMSGrad performs even better than D-PSGD, showing that fixing the divergence issue of DADAM is of significance and supporting the importance of designing theoretically convergent algorithms such as the proposed framework.
>
> We are revising our paper based on all reviewers' comments, we will update you once the revision is done.

---

> > ### Author Response · Authors · 2020-11-24
> > **Revision on linear speedup, convergence measure, and extension to AdaGrad is added**
> >
> > We have revised our paper to further address the reviewer's questions.
> > 1. We revised Theorem 2 and added Corollary 2.1 to show possible linear speedup from the number of nodes. This result only requires changing the stepsize in the last step of our proof.
> > 2. We added a one-sentence comment on the convergence measure below Theorem 3.
> > 3. In section 4, we extended our decentralization framework to AdaGrad to further support the usefulness of the algorithmic converting approach.
> >
> > We hope the reviewer can evaluate the merit of our work based on the main contributions, i.e. the framework for converting adaptive gradient methods into their decentralized counterpart with possible convergence guarantees. The main focus of this work is showing convergence instead of showing benefits over existing algorithms.

---

### Author Response · Authors · 2020-11-24
**Summary of revision**

Based on the reviewers' comments, we have revised the paper to refine and generalize our results. The summary of the revision is listed below.

1. We revised the convergence results so that linear speedup in terms of the number of nodes can be shown.
2. Based on our converting framework, we converted AdaGrad into its decentralized version and analyzed its convergence rate.
3. More discussions and explanations are provided throughout the paper for ease of reading.

---

### Decision · Program_Chairs · 2021-01-07
**Final Decision**

**Decision:**

Reject

**Comment:**

The reviewers have some following concerns:

1) There is lack of experimental result. The experiment on MNIST with small CNN architecture is definitely not sufficient to verify the efficiency of the proposed method. Moreover, the advantage of the proposed method is not very clear due to the choices of the parameters. The choice of the learning rates is quite sensitive.

2) It is not clear why the authors could argue that $ \mathbb{E}(V_T) = \mathcal{O}(T)$ without any theoretical and empirical support. Even if this is correct, this term could dominate the first term unless $ \mathbb{E}(V_T) \leq \mathcal{O}(\sqrt{T})$, which is too strong. If assuming $\mathbb{E}(V_T) = \mathcal{O}(T)$, the convergence results are upper bounded by some constant (note that $\epsilon$ is a constant in this scenario, not arbitrarily small). Hence, the authors failed to show the convergence to a stationary point.

There are some suggestions to improve the paper as follows:

1) Show $\mathbb{E}(V_T) = \mathcal{O}(T)$ and revise the theory properly to make it rigorously by showing upper bounded by some function $R(T) \to 0, T \to \infty$ rather than showing the convergence to some fixed neighborhood. (Note that $\frac{C_4}{\sqrt{N}}$ is a fixed constant).

2) Do more experiments on various datasets and network architectures to verify the efficiency of the proposed method and show the clear advantages compared to others.

3) Provide convergence rate comparisons with other decentralized algorithms (e.g., as a table). It would be nice if the authors also provide the assumptions and the dependent constants so that the readers could really see the differences.

4) Explicitly derive the convergence measure based on the standard one, that is, $\frac{1}{T} \sum_{t=1}^{T} \mathbb{E} [ \| \nabla  f (  X_t )  \|^2 ] $ and add the dependency of $G_{\infty}^2$ to the bound.

5) Revise the paper and implement all necessary comments from the reviewers consistently with the content.